  

EMBO
Molecular Medicine

# Early auto-immune targeting of photoreceptor ribbon synapses in mouse models of multiple sclerosis

Mayur Dembla[1,*] , Ajay Kesharwani[1], Sivaraman Natarajan[1,†], Claudia Fecher-Trost[2], Richard Fairless[3], Sarah K Williams[3], Veit Flockerzi[2], Ricarda Diem[3], Karin Schwarz[1,**] & Frank Schmitz[1,***]

## Abstract

Optic neuritis is one of the first manifestations of multiple sclerosis. Its pathogenesis is incompletely understood, but considered to be initiated by an auto-immune response directed against myelin sheaths of the optic nerve. Here, we demonstrate in two frequently used and well-validated mouse models of optic neuritis that ribbon synapses in the myelin-free retina are targeted by an auto-reactive immune system even before alterations in the optic nerve have developed. The auto-immune response is directed against two adhesion proteins (CASPR1/CNTN1) that are present both in the paranodal region of myelinated nerves as well as at retinal ribbon synapses. This occurs in parallel with altered synaptic vesicle cycling in retinal ribbon synapses and altered visual behavior before the onset of optic nerve demyelination. These findings indicate that early synaptic dysfunctions in the retina contribute to the pathology of optic neuritis in multiple sclerosis.

**Keywords** CASPR1; multiple sclerosis; retina; ribbon synapse; RIBEYE
**Subject Categories** Immunology; Neuroscience

## Introduction

Multiple sclerosis (MS) is a frequent auto-immune inflammatory disease of the central nervous system (CNS). So far, most MS research focused on neuroinflammatory changes within the white matter of the brain, the spinal cord, and the optic nerves with myelin components considered as the primary immune target (Huang *et al*, 2017). Optic neuritis is a frequent manifestation of MS that is primarily attributed to an auto-immune response against components of the myelin sheath of the optic nerve. In the present study, we address changes in the retina, a myelin-free organ in a mouse model of optic neuritis. Except for retinal ganglion cells, the retina has not yet been considered as a primary immune target in MS or optic neuritis, although effects on retinal layers distant from the retinal ganglion cell layer have been suggested by optical coherence tomography, a clinical imaging tool (Behbehani *et al*, 2017; Petzold *et al*, 2017).

Light stimuli perceived by rod and cone photoreceptors are transmitted by ribbon synapses, from the outer retina to the inner retina for further processing, followed by transmission by ganglion cells to the brain. Ganglion cell axons are largely unmyelinated until they leave the eye at the lamina cribrosa to form the optic nerve (Perry & Lund, 1990). Photoreceptor synapses in the outer plexiform layer (OPL) as well as bipolar cell synapses in the inner plexiform layer (IPL) are ribbon-type synapses, continuously active synapses capable of both slow continuous and fast, stimulus-synchronous synaptic vesicle exocytosis (Matthews & Fuchs, 2010). RIBEYE is a unique and essential structural component of synaptic ribbons (Schmitz *et al*, 2000; Maxeiner *et al*, 2016). RIBEYE consists of a characteristic amino-terminal A-domain and a carboxyterminal B-domain (Schmitz *et al*, 2000; Schmitz, 2009). By antibody-based immunopurification of synaptic ribbon protein complexes using a monoclonal antibody against RIBEYE and by subsequent analysis of isolated protein complexes by mass spectrometry, we identified the adhesion proteins CASPR1 and contactin1 (CNTN1) as potential components associated with RIBEYE and the synaptic ribbon complex.

CASPR1 (contactin-associated protein-1; CNTNAP1; Paranodin) is a ≈ 180 kDa multi-domain single-pass transmembrane glycoprotein with sequence similarities to the neurexin family of adhesion proteins (Rasband & Peles, 2015). CASPR1 has been characterized as a component of axoglial junctions in the paranodal region of myelinated axons (Rasband & Peles, 2015). At that site, it forms a cis-complex with contactin1 (CNTN1), a GPI-anchored cell adhesion protein of the immunoglobulin superfamily (Zeng & Sanes, 2017).

1   Department of Neuroanatomy, Institute of Anatomy and Cell Biology, Medical School, Saarland University, Homburg, Germany
2   Institute of Experimental and Clinical Pharmacology and Toxicology, Medical School, Saarland University, Homburg, Germany
3   Department of Neurology, University Clinic Heidelberg, Heidelberg, Germany
   *Corresponding author. Tel: +49 6841 1626195; E-mail: dotdot.mayur@gmail.com
   **Corresponding author. Tel: +49 6841 1626199; E-mail: Karin.Schwarz@uks.eu
   ***Corresponding author. Tel: +49 6841 1626012; E-mail: frank.schmitz@uks.eu
   †Present address: St. Jude Children's Research Hospital, Memphis, TN, USA

The pathophysiological importance of CASPR1/CNTN1 in MS has been attributed to the localization of these proteins in the myelin sheath at the paranodal region of the nodes of Ranvier present along myelinated axons (Coman *et al*, 2006) including the optic nerve (Stojic *et al*, 2018).

In this study, we show that CASPR1 and CNTN1, recently discovered auto-antigens in MS (Stathopoulos *et al*, 2015), are associated with synaptic ribbons, presynaptic specializations in retinal ribbon synapses. We asked whether ribbon synapses could be primary targets of auto-inflammatory changes in MS or optic neuritis even before the beginning of demyelination of the optic nerve. The answer to this question might shed more light on the complex, incompletely understood pathogenesis of optic neuritis and introduces the retina as an early affected organ despite its lack of myelin.

## Results

### CASPR1 and synaptic ribbons co-immunopurify

We used a newly generated, mouse monoclonal antibody against RIBEYE(B)-domain/CtBP2 to immuno-isolate synaptic ribbon complexes from bovine retina (Fig 1). The specificity of this mono-clonal antibody was verified by Western blot of proteins from wild-type and RIBEYE knockout mice (Appendix Fig S2C). Irrelevant mouse immunoglobulins served as negative controls for the immunoprecipitation (IP; Fig 1A). Among the Coomassie-stained proteins analyzed by SDS–PAGE, we observed a strong enrichment of a protein of 120 kDa that was absent in the control IP (Fig 1A). The Western blot performed in parallel confirmed that the 120 kDa protein represents RIBEYE (Fig 1Ba).

Another prominent protein of 180 kDa was present in the RIBEYE but not in the control IP (Fig 1A). This 180 kDa protein was unambiguously identified by mass spectrometry as CASPR1 (24% amino acid sequence coverage; Appendix Fig S1). Western blots of the immunopurified synaptic ribbon complexes further confirmed specific enrichment of CASPR1 (and RIBEYE) in the RIBEYE but not in the control IP (Fig 1Bb). Compared to RIBEYE, CASPR1 was less strongly enriched indicating that only part of the CASPR1 protein is associated with ribbons. Among the ≈120 kDa proteins immunopre-cipitated with RIBEYE antibodies was an additional protein, contact-in1 (CNTN1, Fig 1A), as proved by mass spectrometry (30% sequence coverage, Appendix Fig S1), and Western blot (Fig 1Bc). In Fig 1Bd, heavy chains of the precipitating antibody (IgG hc) were visualized as loading control.

### CASPR1 is synaptically enriched close to the synaptic ribbon in photoreceptor ribbon synapses

The retina is largely devoid of myelinated axons, and therefore, the observed co-immunoprecipitation of both CASPR1 and CNTN1 with synaptic ribbons was surprising. To validate this result, we employed antibodies for immunolocalization of CASPR1 in the retina. The three independent antibodies directed against different epitopes of CASPR1 detected the same single 180 kDa CASPR1 protein in Western blot (Appendix Figs S2A and S10C). In the following, we used these antibodies for morphological analyses

predominantly on 0.5-μm-thin resin sections (Figs 2 and 3) according to Wahl *et al* (2016) and Eich *et al* (2017) which allowed a much better resolution than immunolabeling of 10-μm-thick cryostat sections (Appendix Fig S3). The third independent antibody against CASPR1 (5F9 mouse monoclonal antibody) was applied on cryostat sections because it did not work on semi-thin resin sections (Appendix Fig S10D, E, and G). All CASPR antibodies produced a very similar synaptic immunosignal of CASPR1 that was highly enriched at the synaptic ribbon irrespective of the applied method (Fig 2A1 and A3; see also Fig 7A1 and A3, Appendix Figs S3–S11).

Using confocal immunofluorescence microscopy with the anti-body against the aminoterminus of CASPR1, we observed a strong enrichment of CASPR1 immunosignals in both synaptic layers of the retina, in the outer and inner plexiform layer (Figs 2 and 3; Appendix Fig S6A). Particularly, the photoreceptor ribbon synapses in the OPL showed a strong CASPR1-specific immunoreactivity in close vicinity to the synaptic ribbon as judged by high-resolution confocal imaging and super-resolution structured illumination microscopy (SR-SIM; Appendix Fig S6B and C). The synaptic CASPR1 immunosignal was absent after pre-incubation of the CASPR1 antibody with the corresponding peptide antigen, but not after pre-absorption with an unrelated peptide (Appendix Fig S4). Similar results were also observed with the second CASPR1 antibody that is directed against the intracellular carboxyterminus of CASPR1 (Appendix Figs S3B and C, S4B, and S5A–C) and the newly generated monoclonal antibody against CASPR1 (Appendix Fig S10).

To further characterize CASPR1 localization, we performed triple-immunolabeling analyses, with antibodies against the synaptic marker proteins PSD95, CASK, and RIMs (Appendix Figs S7 and S8). PSD95 is a presynaptic scaffold protein in photoreceptor synapses and antibodies against PSD95 immunolabel virtually the entire plasma membrane of the presynaptic terminal (Maxeiner *et al*, 2016). SR-SIM analyses of triple-immunolabeled single synapses (Appendix Fig S8A) and high-resolution confocal analyses (Appendix Fig S7A and B) indicated that the majority of the CASPR1 immunosignal is located within the PSD95-immunolabeled presy-naptic terminal suggesting a predominant presynaptic localization of CASPR1 in photoreceptor ribbon synapses. A partly presynaptic localization of CASPR1 was further supported by co-immunola-beling analyses with antibodies against CASK and RIM (Südhof, 2012), two active zone proteins in retinal synapses (Schmitz *et al*, 2000; Anjum *et al*, 2014; Appendix Fig S8B–E). These experiments demonstrated significant overlap of CASPR1 with the presynaptic active zone markers RIMs/CASK indicating a localization of CASPR1 in close vicinity to the active zone (Appendix Fig S8). In agreement with the proposed presynaptic localization, the bulk of synaptic CASPR1 did not co-localize with mGluR6 (Appendix Fig S8F) which is located postsynaptically at the tips of invaginating ON-bipolar cells (Katiyar *et al*, 2015). Some CASPR1 immunoreactivity was also found in some distance to the active zone (Appendix Fig S3B and C). This extrasynaptic CASPR1 immunosignal probably represents CASPR1 localized to the neuropil region of the OPL, as previously demonstrated (O'Brien *et al*, 2010).

Like CASPR1, also CNTN1 was highly enriched in the OPL of the mouse retina (Figs 1B and 3A; Appendix Figs S5C, S9A–C, S11C and G, and S13A) and also present in non-synaptic layers (Fig 3A). In the IPL, we observed a lower amount of immunoreactive CNTN1 than in the OPL (Fig 3A).

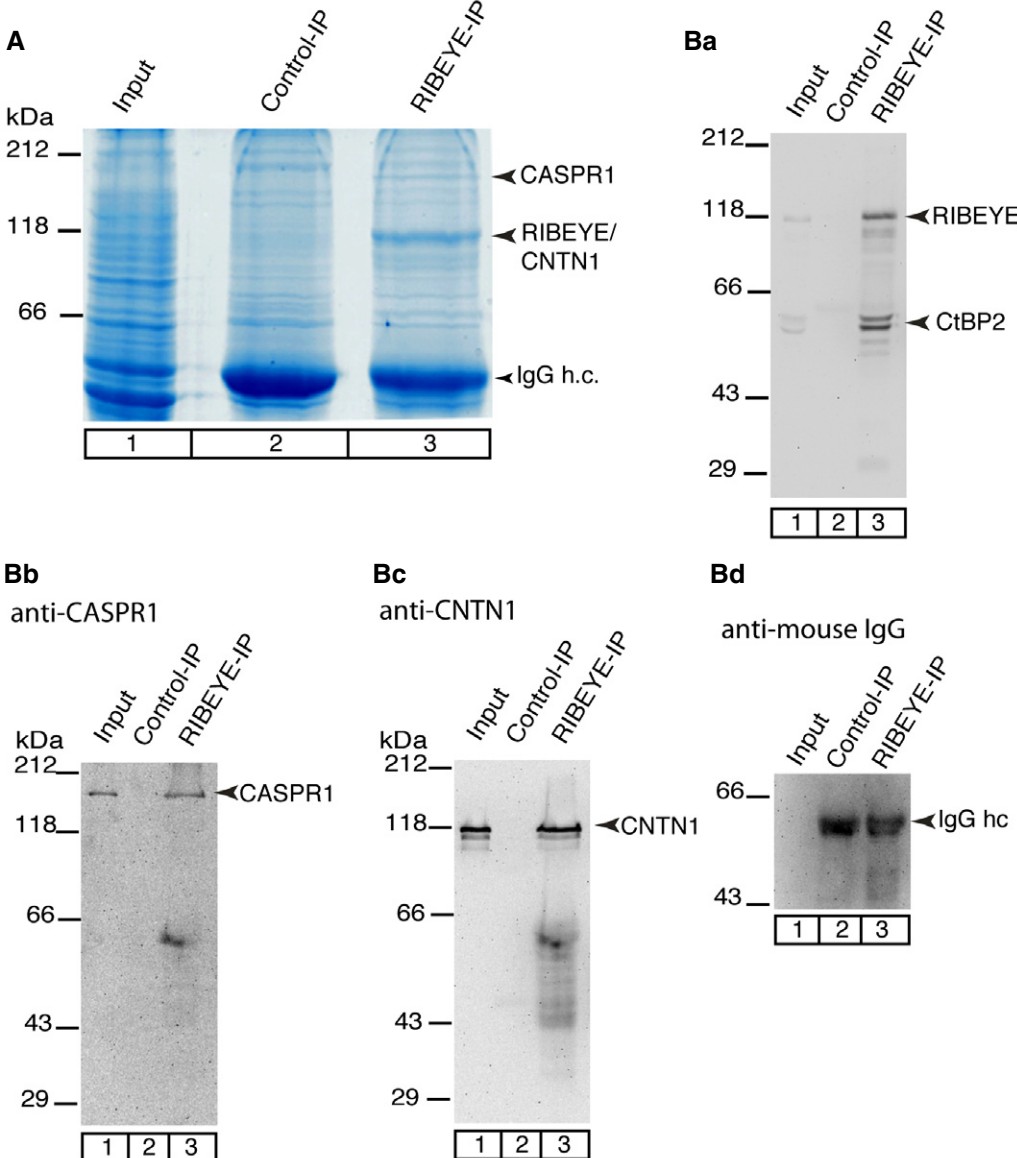

**Figure 1.  CASPR1 and CNTN1 co-immunoprecipitate with RIBEYE.**

A   Coomassie-stained proteins from bovine retina that were co-immunoprecipitated by the anti-RIBEYE antibody 2D9 (lane 3). Lane 2, control precipitation by irrelevant mouse IgGs; lane 1, input (10% of the total loaded in each lane).

B   Western blot of the immunoprecipitated proteins (as in A, 1% of the total protein was loaded per lane) probed with anti-RIBEYE (Ba), anti-CASPR1 (Bb), and anti-CNTN1 (Bc). The 2D9 antibody immunoprecipitated RIBEYE (Ba, lane 3), CASPR1 (Bb, lane 3), and CNTN1 (Bc, lane 3); these proteins were not detectable in the control immunoprecipitations. (Bd) As a loading control, heavy chains of the precipitating antibody (IgG hc) were visualized by the goat anti-mouse IgG antibody conjugated to peroxidase.

Source data are available online for this figure.

The ultrastructural localization of CASPR1 and CNTN1 was resolved by pre- and post-embedding immunogold electron microscopy (Fig EV1). CASPR1 localization was solved by pre-embedding immunogold electron microscopy (Fig EV1B1–B3); CNTN1 localization by post-embedding immunogold electron microscopy (Fig EV1A1–A3). In photoreceptor ribbon synapses, a significant portion of both CASPR1 and CNTN1 was associated with the synaptic ribbon complex. Pre-embedding immunogold electron microscopy, that preserves membrane contrast, suggested that CASPR1 is present on ribbon-associated synaptic vesicles (arrows in Fig EV1B1). A portion of CASPR1/CNTN1 was also associated with the presynaptic plasma membrane close to the active zone (Fig EV1A1, A2, and B2). A portion of CASPR1/CNTN1 was also present on the postsynaptic tips of horizontal cells located directly opposite to the active zone (Fig EV1A2, B2 and B3).

    

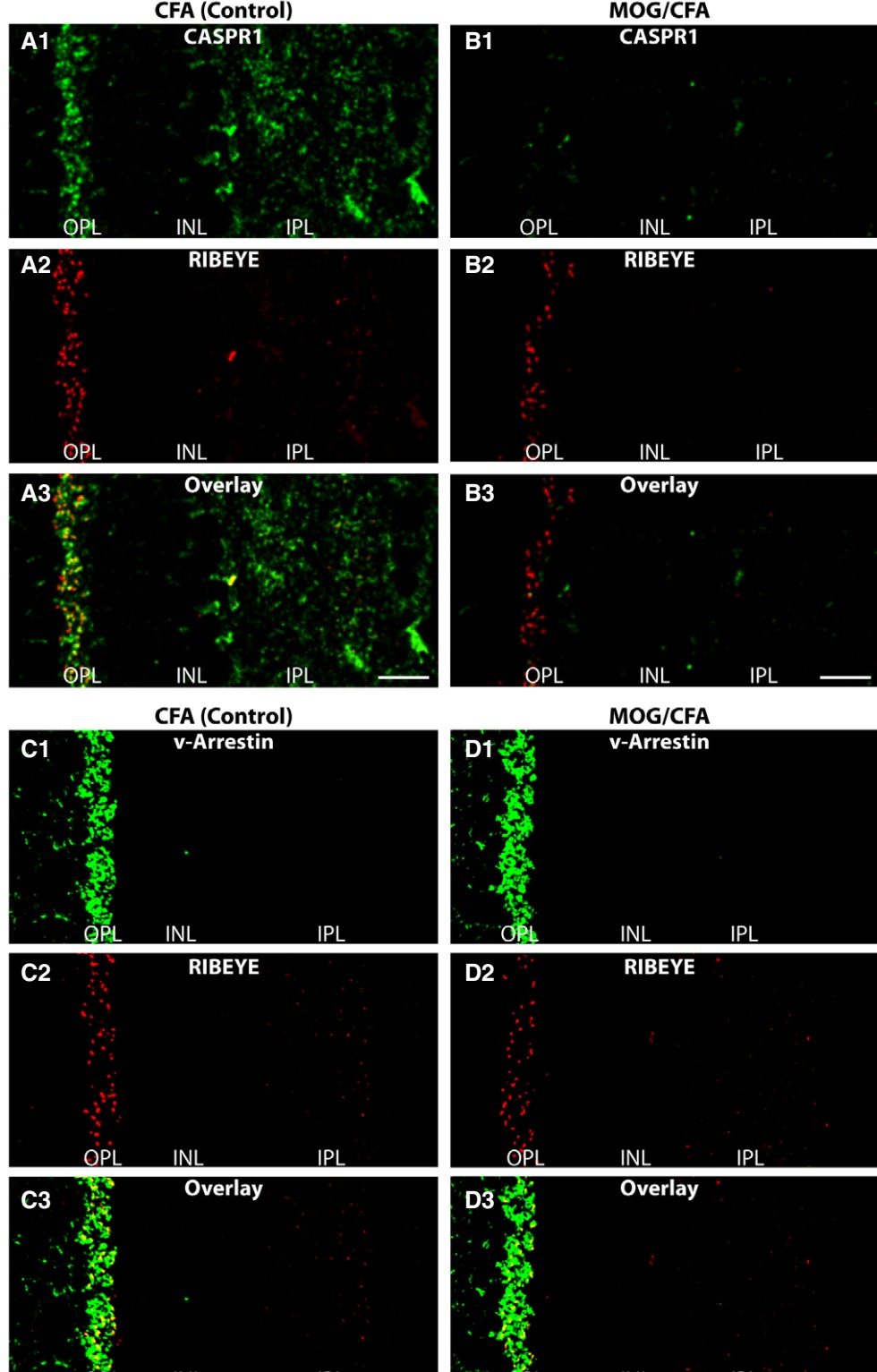

**Figure 2. Localization of CASPR1 and visual arrestin (v-Arrestin) in semi-thin sections of retinas from either MOG/CFA-injected mice (EAE mice) or CFA-injected control mice (9 days after injection).**

A–D   Semi-thin sections were double-immunolabeled with antibodies against CASPR1 and RIBEYE (A, B) or with antibodies against v-Arrestin and RIBEYE (C, D). For quantification of the immunolabeling data, please see Fig 3C. v-Arrestin, visual arrestin; OPL, outer plexiform layer; INL, nuclear layer; IPL, inner plexiform layer. Scale bars: 20 μm.

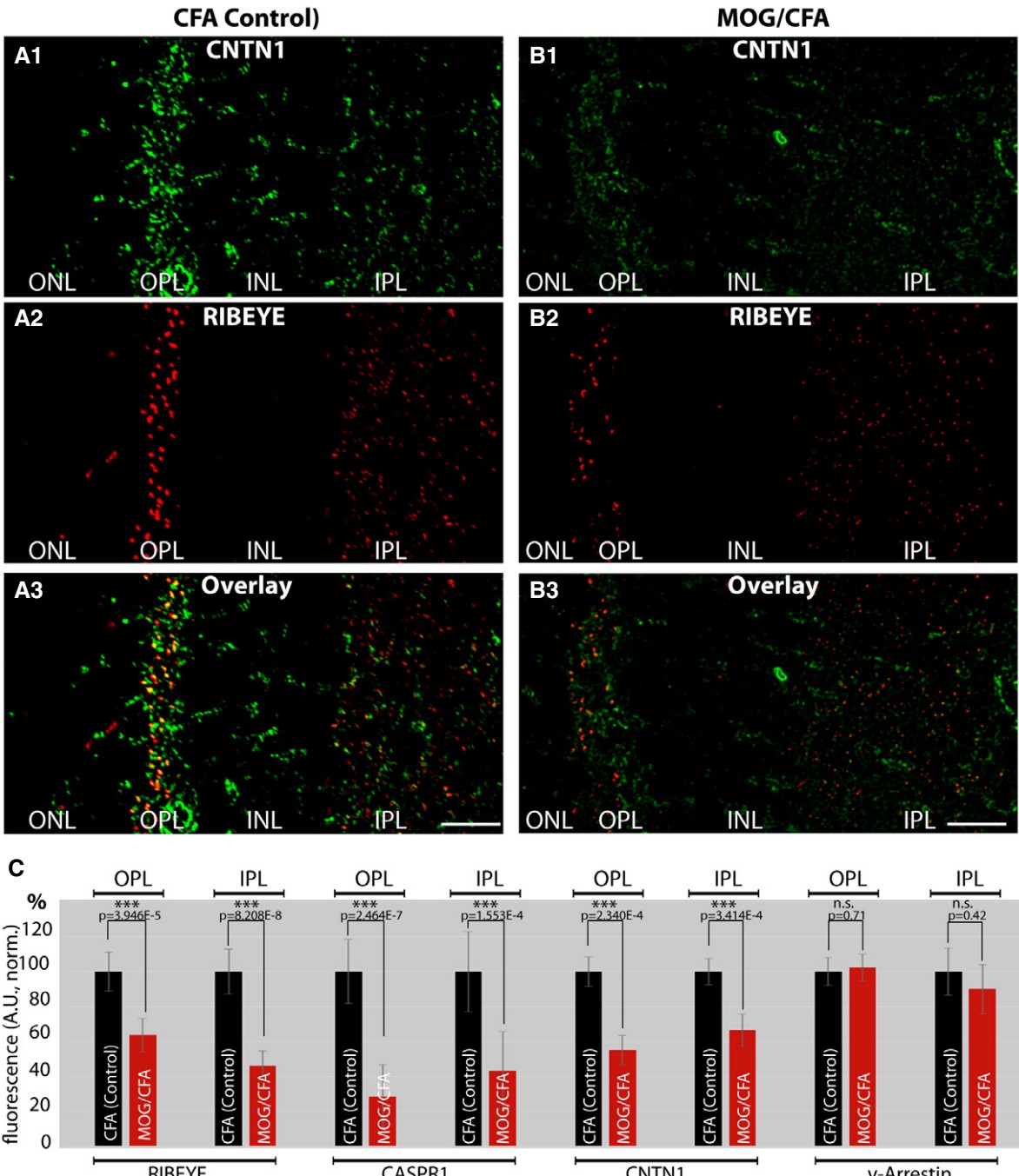

**Figure 3. Localization of CNTN1 in semi-thin (0.5-μm-thin) sections of retinas from either MOG/CFA-injected mice (EAE mice) or CFA-injected control mice (9 days after injection).**

A, B   Sections were double-immunolabeled with antibodies against CNTN1 and RIBEYE. Scale bars: 20 μm.

C   Quantification of the immunosignals of the indicated proteins in the OPL and IPL. ONL, outer nuclear layer; OPL, outer plexiform layer; INL, nuclear layer; IPL, inner plexiform layer; n.s., non-significant; ***$P < 0.001$ (Mann–Whitney $U$-test, actual $P$-values are given in the figure) $N = 6$ independent set of experiments; $n = 295$ images (CFA); $n = 274$ images (MOG/CFA) for quantification of RIBEYE immunosignals; $n = 95$ (CFA); $n = 92$ (MOG/CFA) for quantification of CASPR1 immunosignals; $n = 91$ (CFA), $n = 82$ (MOG/CFA) for CNTN1 quantification; $n = 109$ (CFA); $n = 100$ (MOG/CFA) for quantification of v-Arrestin. Error bars are ± SEM.

## Synaptic CASPR1 expression in the retina is severely altered in the early induction phase of optic neuritis

As outlined above, CASPR1 and CNTN1 are frequent auto-antigens in auto-immune diseases including multiple sclerosis (Stathopoulos et al, 2015). Since we found a portion of CASPR1 and CNTN1 to be associated with retinal ribbon synapses in close vicinity to the synaptic ribbons particularly in photoreceptor ribbon synapses, we asked whether this synaptic CASPR1/CNTN1 complex might be a neuroinflammatory target in mouse models of optic neuritis/MS.

**Figure 4. MOG/CFA-injected but not CFA-injected mice contain reactive antibodies for retinal proteins in blood and show altered spatial visual behavior.**

A    Seven to nine days after injection, blood from MOG/CFA-injected (red) but not from CFA-injected mice (black), elicited strong antibody responses in ELISA. Antibodies for RIBEYE and SV2 were used as positive controls (green), secondary antibody only as a negative control (white). Normalized data are given as mean $\pm$ SEM [$N$ = 4 (CFA); $N$ = 5 (MOG/CFA)]. **$P$ < 0.01 (significance was assessed by Mann–Whitney $U$-test, precise $P$-values given in the figure).

B    Control Western blot of HEK293 protein lysates obtained from cells transfected with mCherry (lane 1) or CASPR1-mCherry cDNA (lane 2) and probed with anti-mCherry antibody.

C, D    Western blot of protein lysates as in (B) probed by blood samples from mice before and 7–9 days after MOG/CFA injection. Only blood samples obtained from MOG/CFA-injected mice revealed immunoreactivity against CASPR1 (lanes 15, 16; top panels), whereas samples obtained before immunization (lanes 7, 8; top panels) did not. The same filters were re-probed with anti-actin antibodies as loading control.

E    Summary of data from (C, D) shown as mean $\pm$ SEM ($N$ = 4, before injection; $N$ = 5, MOG/CFA-injected). ***$P$ < 0.001; n.s., non-significant (unpaired two-tailed Student's $t$-test, precise $P$-values given in the figure).

F    Spatially guided visual behavior, measured as frequency threshold with a virtual optokinetic system 9 days after immunization, is significantly decreased in MOG/CFA-injected mice compared to CFA-injected control litters. Data are shown as median; boxes and whiskers illustrate 25–75 and 5–95 percentiles of values, respectively. Points below and above whiskers are drawn as individual points. At each trial, mice were measured three times. Home-made and commercial MOG/CFA suspensions were compared as well as PLP-injected mice with control-injected mice. Statistical analysis was done by ANOVA (0.05 significance level), and mean comparison probabilities were Bonferroni corrected. $N$ = 32, $n$ = 96 (MOG/CFA); $N$ = 24, $n$ = 72 (CFA). ***$P$ < 0.001 (Mann–Whitney $U$-test). Results from mice injected with home-made and commercial MOG/CFA suspensions as well as results obtained from PLP/CFA-injected mice (and respective control-injected mice) are displayed.

Source data are available online for this figure.

First, we analyzed EAE mice obtained by injection of the myelin oligodendrocyte glycoprotein (MOG) peptide (in CFA), as described in "Materials and Methods". We analyzed retinas from female mice in the pre-clinical phase of disease induction, i.e., 9 days after injection of the MOG peptide. We used both home-made MOG/CFA suspensions (Figs 2–5 and EV3–EV5) as well as commercially available MOG/CFA suspensions (Figs 6 and EV2; Appendix Figs S11, S12 and S14) to induce EAE. This optic neuritis model is a subtype of experimental auto-immune encephalomyelitis (EAE; Williams *et al*, 2011). The early time point was selected to exclude secondary changes resulting from demyelination or from a loss of retinal ganglion cells.

As demonstrated in Figs 2 and 3, we found a strongly decreased synaptic immunoreactivity for both CASPR1 (Fig 2B1 and B3) and CNTN1 (Fig 3B1 and B3) in sections from MOG/CFA-injected mice in comparison with sections from complete Freund's adjuvant (CFA)-injected controls (Figs 2A1 and A3, and 3A1 and A3; quantification in Fig 3C). In contrast, the immunoreactivity of visual arrestin was not changed (Fig 2D1 and D3) in MOG/CFA-injected mice in comparison with CFA(control)-injected mice indicating (Fig 2C1 and C3) that the changes are specific to CASPR1/CNTN1 and not a result of global changes of protein amounts (quantification in Fig 3C). We also noted that the synaptic ribbons as visualized by antibodies against RIBEYE showed a decreased immunoreactivity in MOG/CFA-injected mice (Figs 2B2, B3, D2 and D3, and 3B2 and B3, quantification in Fig 3C) in comparison with CFA-injected control mice (Figs 2A2, A3, C2 and C3, and 3A2 and A3, quantification in Fig 3C). The decrease in the amount of CASPR1 and RIBEYE proteins in MOG/CFA-injected mice in comparison with controls mice was also observed in Western blots (Fig EV5). The results described above were obtained with home-made MOG/CFA peptide suspensions. Virtually identical results were obtained with MOG/CFA suspensions from a commercially available kit (Hooke kit; Appendix Fig S11) demonstrating that the synaptic decrease in CASPR1 and CNTN1 does not depend on the source of MOG/CFA peptide suspension.

## Massive increase in auto-reactive anti-retinal antibodies in early pre-clinical optic neuritis

In order to address the reason for the increased recruitment of complement proteins to retinal synapses and for its activation during the early pre-clinical stage of optic neuritis/EAE, we collected blood samples from MOG/CFA-injected and CFA(control)-injected mice (in the pre-clinical phase, 7–9 days after immunization) and tested them in ELISA assays for the presence of auto-reactive antibodies against retinal proteins. For this purpose, we absorbed mouse retinal lysates to ELISA plates and probed them with the blood samples from MOG/CFA-injected mice and CFA-injected control mice. Blood samples were obtained both before immunization either with MOG/CFA or CFA only as well as on days 7–9 after injection with MOG/CFA or CFA (control). The antibody titers of the serum samples obtained on days 7–9 after immunization were normalized to antibody titers obtained from serum samples before injection.

In these experiments, we observed a strong increase in antibody titer directed against retinal proteins in MOG/CFA-injected mice 7–9 days after injection in comparison to CFA-injected control mice (Fig 4A). Remarkably, despite the short time window of 7–9 days, the immune response of the MOG/CFA-immunized animals was similarly strong as our positive controls in which we determined the reactivity of the absorbed retinal lysate with antibodies against RIBEYE (2D9) and SV2 (Fig 4A, green bars).

The ELISA results in Fig 4A demonstrated a strong antibody auto-immunoreactivity against retinal proteins without revealing the identity of the antigens against which the auto-immune response was directed. To analyze whether the auto-reactive immune response at day 7–9 after injection includes antibodies that specifically target CASPR1, we heterologously expressed full-length CASPR1 cDNA to obtain a mCherry-tagged fusion protein in HEK293 cells (Fig 4B). HEK293 cells transfected with the mCherry cDNA alone (Fig 4B) served as negative controls. We tested serum samples obtained shortly before immunization as well as samples obtained at days 7–9 after immunization with either MOG/CFA or CFA alone (control; Fig 4C–E).

A strong immunoreactivity against CASPR1 was detected in sera from MOG/CFA-injected mice using this *in vitro* system (Fig 4D and E). Sera from MOG/CFA-injected mice did not react with lysates from control-transfected HEK293 cells (Fig 4D and E). CASPR1 was immunodetected only by serum from MOG/CFA-immunized mice (Fig 4D and E) but not by CFA-injected control mice (Fig 4D and E) and also not by all serum samples obtained before injection (Fig 4C

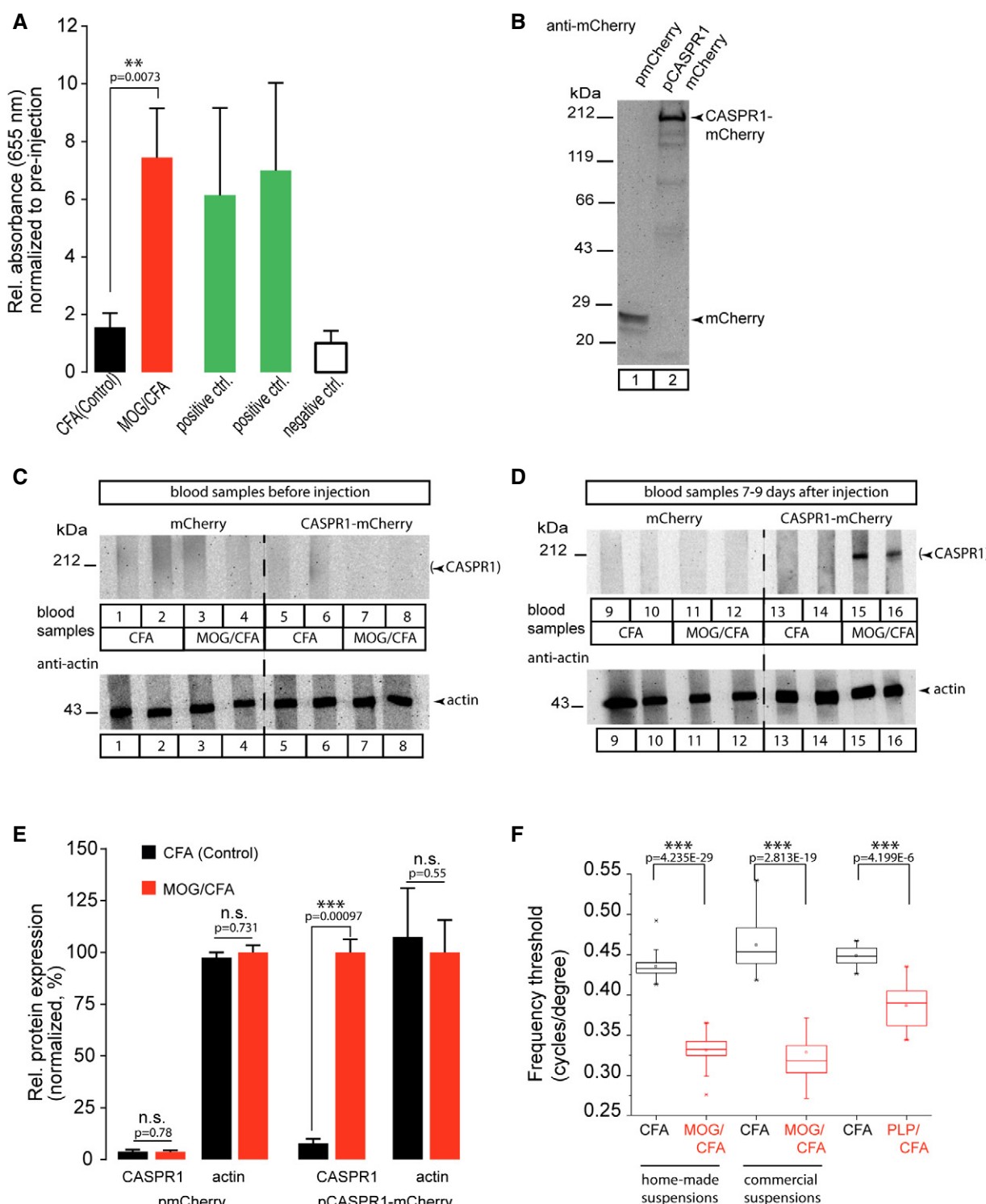

**Figure 4.**

and E). These results show that the MOG/CFA injection specifically induces a strong auto-reactive antibody-mediated immune response against CASPR1. The results described above were obtained with self-made MOG/CFA peptide suspensions. Virtually identical results were obtained with MOG/CFA suspensions from a commercially available kit (Fig EV2). These results demonstrate that the induction

of auto-antibodies against CASPR1 is independent of the source of MOG/CFA peptide suspension.

The auto-antibodies against CASPR1 were not only found in blood samples but also in the cerebrospinal fluid (Fig EV2A) indicating that the antibodies have crossed the blood–brain barrier and were thus able to target synapses. This view was supported by the

**Figure 5. Optic nerve parameters are unaltered in early EAE (9 days after injection).**

A, B    Cryostat sections of the optic nerve from MOG/CFA-injected mice (own lab-made suspension; for comparison with commercial MOG/CFA suspensions, see Appendix Fig S12) and CFA-injected control mice were stained with anti-Na$_v$ antibody (1:50 dilution) to label the node of Ranvier and with anti-CASPR1 antibody (1:500 dilution) to stain the paranodal region.

C, D    Length of the node of Ranvier and length of the paranodal region were quantified.

E, F    (E1, F1) Richardson Blue-stained semi-thin sections from the indicated optic nerve samples. (E2, E3, F2, F3) Transmission electron micrographs of the optic nerves from MOG/CFA- and CFA-injected animals. ax, axon; my, myelin sheath.

G, H    Axon number and myelin thickness were quantified.

Data information: Scale bars: 1 µm (A, B, E, F). Error bars are ± SEM; statistical test: Mann–Whitney *U*-test (Origin Pro).

finding that the complement system was strongly recruited to retinal synapses at this time point, i.e., 9 days after immunization. Complement protein 3 (C3) is a key protein of the complement cascade, both in the classical, alternative, and lectin pathway. We found that C3 protein is already present at retinal synapses of control-injected CFA mice (Fig EV3A1 and C). Similar results to those observed in CFA-injected control mice (Fig EV3A) were also obtained in non-injected wild-type mice (data not shown) demonstrating that complement protein C3 is already present at retinal synapses under control conditions. Remarkably, the amount of C3 protein localized at retinal synapses was significantly enhanced in retinas of MOG/CFA-injected mice in comparison with retinas of CFA-injected control mice, 9 days after injection (Fig EV3B1 and C). The presence of the full-length C3 protein in CFA controls and MOG retinas was confirmed by Western blot with anti-C3 antibodies (Fig EV3D). In Western blots of retinal lysates from MOG/CFA-injected mice, we found an increased amount of C3 full-length protein as well as cleaved C3c protein that could indicate activation of the complement system (Fig EV3D). The total amount of full-length C3 protein in retinal lysates was found to be not significantly changed in Western blots (Fig EV3E) indicating that increased C3 protein at retinal synapses results from increased recruitment from other intraretinal sites. The complement system could be activated by the classical complement pathway by the anti-CASPR1 auto-antibodies that target retinal ribbon synapses.

In order to further corroborate an early pre-clinical activation of the focal complement system, we made use of a monoclonal C5b-9 antibody that is directed against a conformational neo-epitope of the activated complement complex. C5b-9 immunoreactivity of the activated complement complex was significantly increased in retinas of MOG/CFA-immunized mice compared to CFA-immunized control mice (Fig EV4A–C and G). In three out of five MOG/CFA-injected mice (obtained 9 days after immunization), we observed a strong C5b-9 immunoreactivity of the activated complement complex in retinal synapses but not in any of the CFA control-injected mice (Fig EV4A–C and G). The C5b-9 immunosignals of the activated complement complex overlapped with the immunosignals of complement protein C3 at retinal synapses (Fig EV4D–F). The results from these experiments indicate that complement proteins are not only recruited to retinal synapses in MOG/CFA-injected mice vs. CFA-injected controls but that complement proteins are also activated at retinal synapses at that early time point in the pre-clinical phase of EAE/optic neuritis.

**Synaptic vesicle cycling is severely compromised in rod photoreceptor ribbon synapses in early pre-clinical optic neuritis**

Next, we asked whether the retina synapses, that we showed to be targeted by an auto-reactive immune system in the EAE mouse model,

are still functionally active or compromised at that stage. We focused on photoreceptor ribbon synapses because these were particularly sensitive and the most affected synapses (Figs 3C and 7C, and Appendix Figs S11 and S15). To address this question at the cellular level, we generated a transgenic mouse line in which the SypHy protein, a fusion protein of synaptophysin with pH-sensitive GFP (Dreosti & Lagnado, 2011; Linares-Clemente *et al*, 2015), is expressed under the control of the mouse rod opsin promotor to report synaptic vesicle cycling in mouse rod photoreceptor ribbon synapses (Fig 6).

In the control mice, we found robust responses in rod photoreceptor ribbon synapses in response to depolarization and robust recovery after removal of the depolarizing stimulus (Fig 6A). The release rate during depolarization could be fitted best by a double-exponential curve (Fig 6B). In contrast to the control-injected mice, we found a strong decrease of synaptic vesicle cycling in the MOG-/CFA-injected mice. Particularly the amplitude of release as measured by an increase in SypHy fluorescence (fast and slow component of release) as well as the kinetics of release were compromised (Fig 6C). These data show that photoreceptor rod synapses are still alive but functionally impaired at that stage.

In line with these observations, we also found altered visual behavior in MOG/CFA-injected mice in comparison with CFA-injected control mice using a virtual optokinetic reflex tracking system (Fig 4F). Optokinetic tracking is a highly sensitive readout of visual function and strongly dependent on photoreceptor function and synaptic transmission in the retina (Alam *et al*, 2015; Sarria *et al*, 2015). In parallel with the early synaptic alterations, we found a significantly diminished performance in spatial visual behavior, as determined by a decreased frequency threshold (Fig 4F). Virtually identical results were observed in mice immunized either with the home-made MOG/CFA suspensions or with the commercial MOG/CFA suspensions (Fig 4F) demonstrating that the source of the MOG/CFA suspension does not affect the outcome of the visual behavior experiments.

Remarkably, this drop of synaptic performance as measured by SypHy imaging (Fig 6A–C) and by visual behavior (Fig 4F) happened before morphological alterations in the optic nerve were detectable (Figs 5 and 7; Appendix Fig S12). We found that the optic nerve did not show signs of optic neuritis at the time when synaptic dysfunctions were already present. This was shown by various analyses of the optic nerve sourced from the same mice from which the retinas were assayed for synaptic alterations. Stojic *et al* (2018) found that the paranodal region of the optic nerve is an early and sensitive marker for optic neuritis. Therefore, we screened the organization of the paranodal region of the optic nerve at day 9 after immunization by immunolabeling. We applied antibodies against Nav-channels to label the node of Ranvier and antibodies against CASPR1 that label the paranodal region at the node of Ranvier

**Figure 5.**

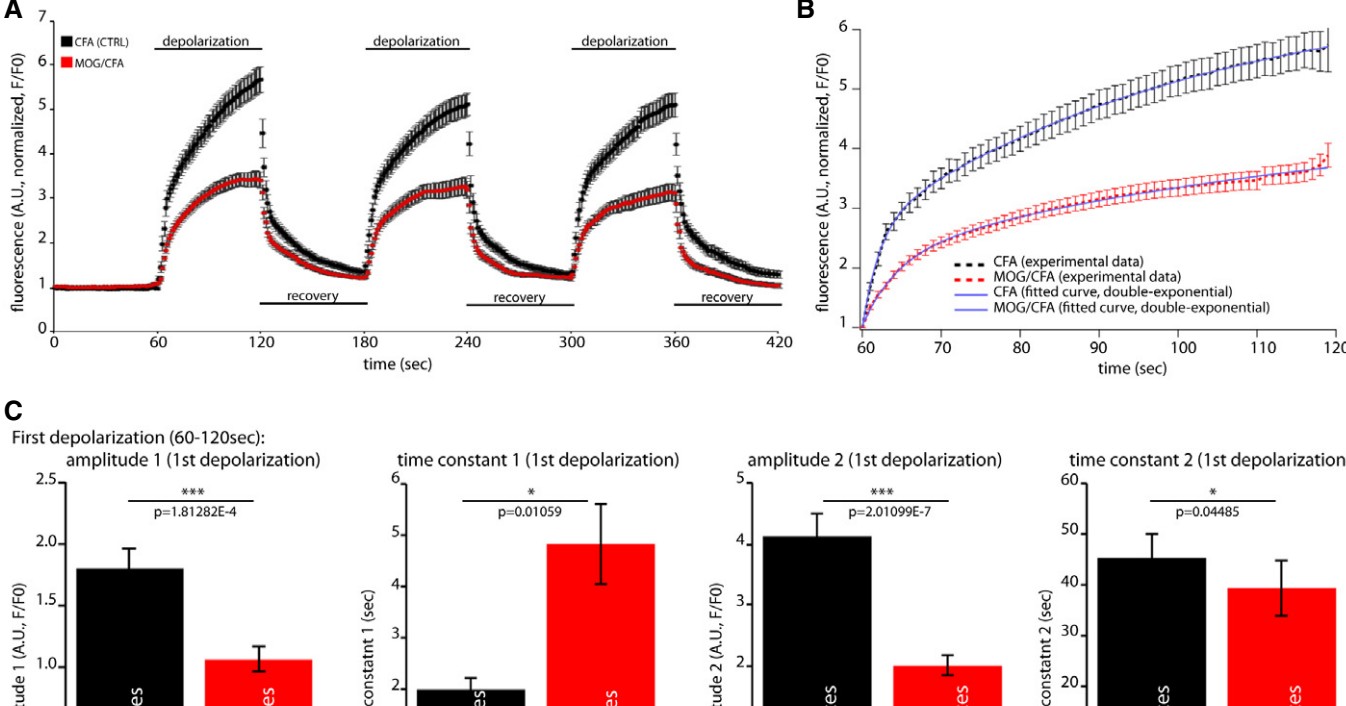

**Figure 6.    Vesicle cycling in MOG/CFA-injected mice (injected with commercial MOG/CFA suspensions; Hooke kit) at day 9 after injection is strongly reduced in comparison to control mice.**

A–C    Retinal slices from MOG/CFA-injected and CFA-injected SypHy mice were stimulated by applying a 25 mM K$^+$-containing depolarization solution for 1 min (A). In response to depolarization, we observed a strong increase in fluorescence (A) that can be best fitted by a double-exponential curve (B). $N$ = 5 (for each CFA and MOG/CFA), $n$ = 35 slices (CFA), $n$ = 46 slices (MOG/CFA). Amplitudes of fast and slow release (together with the respective time constants) of the first depolarization response are plotted in (C). Error bars are ± SEM; statistical test: Mann–Whitney $U$-test (Origin Pro).

(Fig 5A1–3 and B1–3). The length of the nodes of Ranvier and the paranodal regions were quantitatively evaluated (Fig 5C and D) revealing that the Nav-channel-positive nodal and CASPR1-positive paranodal regions were completely unchanged. The length of the Nav-channel-labeled node of Ranvier and the CASPR1 labeled paranodal region were indistinguishable from the control mice (Fig 5C and D). Similarly, also the number of axons in the optic nerve and the thickness of the myelin sheath, as shown by light (Fig 5E1 and F1) and electron microscopic analyses (Fig 5E2, E3, F2 and F3), were unchanged demonstrating that no demyelination or axon loss had taken place at that time point (Fig 5E–H). Quantification of the

number of axons in the optic nerve of MOG/CFA- and CFA(control)-injected mice is given in Fig 5G; the quantification of myelin sheath thickness is presented in Fig 5H.

Remarkably, the described phenomena, i.e., generation of auto-antibodies against CASPR1, and decrease in synaptic expression of CASPR1 and CNTN1 in retinal synapses, are not only common to the MOG/CFA-induced EAE mouse model but occurred in a very similar manner also in the proteolipid protein (PLP) mouse model of multiple sclerosis (Fig 7). In the PLP mouse model, EAE is induced by injection with a peptide from the proteolipid protein (PLP) in CFA. Also in the PLP mouse model of multiple sclerosis/

**Figure 7.    The amount of CASPR1 protein in the synapse is also altered in the PLP mouse model of multiple sclerosis.**

A, B    Semi-thin sections of PLP/CFA- and CFA-injected mice were probed with the indicated antibodies (see also Appendix Fig S13).

C    The synaptic amount of CASPR1 is strongly reduced both in the OPL and IPL. RIBEYE is only reduced in the OPL but not significantly in the IPL. CNTN1 is strongly reduced in both OPL and IPL (see also Appendix Fig S13). $N$ = 4 PLP/CFA-injected mice; $N$ = 4 CFA-injected mice; $n$ = 140 slices (CFA, RIBEYE); $n$ = 143 slices (PLP/CFA, RIBEYE); $n$ = 47 slices (CFA, CASPR1); $n$ = 53 slices (PLP/CFA, CASPR1); $n$ = 49 slices (CFA, CNTN1); $n$ = 46 slices (PLP/CFA, CNTN1); $n$ = 44 slices (CFA, v-Arrestin); $n$ = 44 slices (PLP/CFA, v-Arrestin).

D, E    Cryostat sections of the optic nerve from PLP and control mice were stained with anti-Nav-channel antibody to label the node of Ranvier and with anti-CASPR1 antibody to stain the paranodal region.

F    Transmission electron micrographs of the optic nerves from PLP/CFA- and CFA-injected animals (F1–F4).

G, H    Length of the node of Ranvier and length of the paranodal region were quantified.

I, J    Axon numbers and myelin thickness were quantified.

    

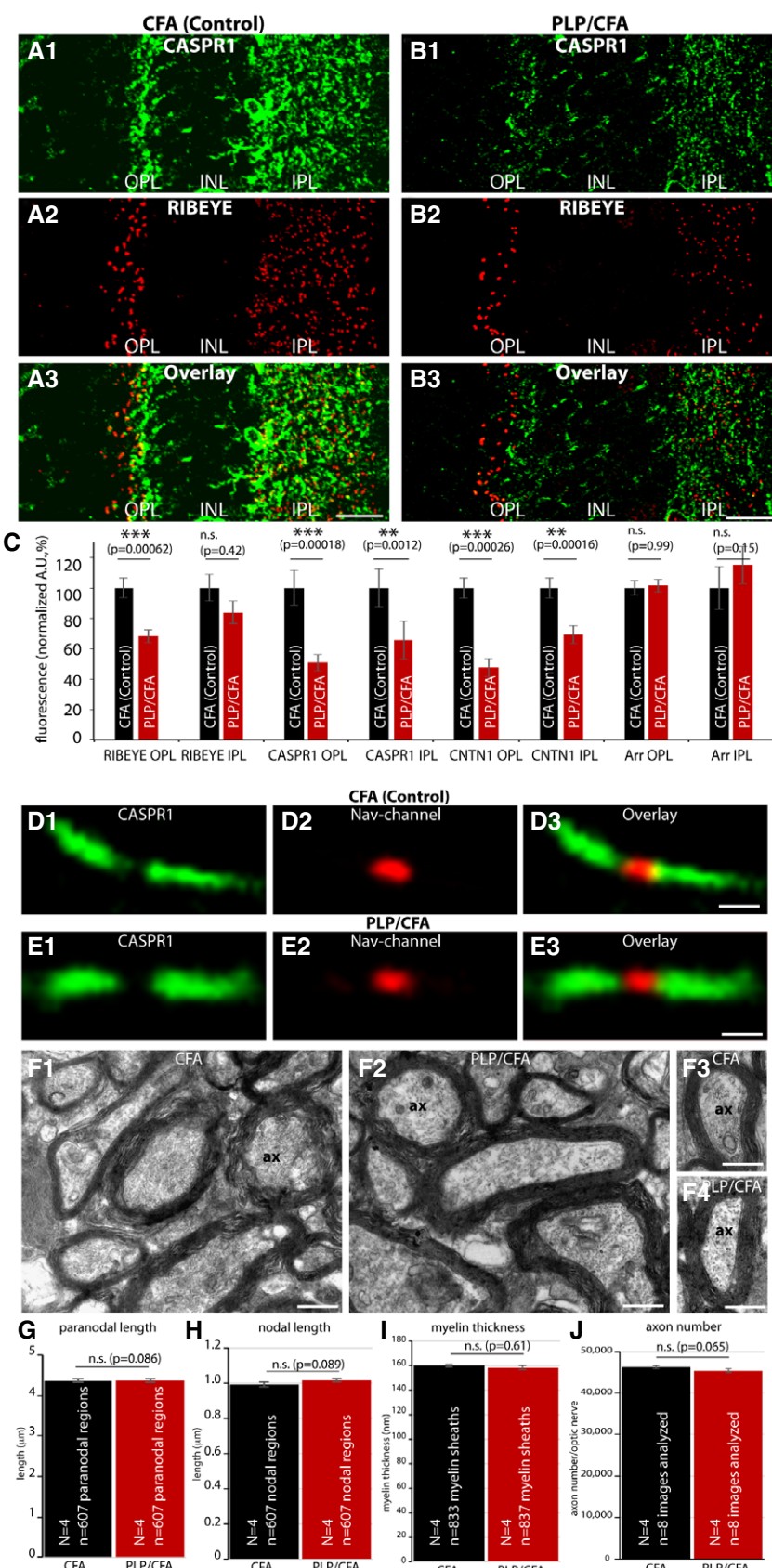

**Figure 7.**

optic neuritis, we observed a decrease in CASPR1 and CNTN1 in retinal synapses (Fig 7A1–3 and B1–3; Appendix Fig S13A1–3 and B1–3). Only the effect on RIBEYE expression appeared slightly smaller (Fig 7C). There was a strong decrease in RIBEYE in the outer plexiform layer (OPL) without a significant decrease in the inner plexiform layer (Fig 7). Visual arrestin was unchanged in expression in PLP/CFA-injected mice in comparison with CFA(control)-injected mice (Appendix Fig S13C1–3 and D1–3). Remarkably, also in the PLP mouse model, the organization of the nodal and paranodal regions, as visualized by immunolabeling with antibodies against Nav-channels and antibodies against CASPR1, was unchanged at that time point (Fig 7D1–3 and E1–3; for quantification, see Fig 7G and H). Similarly, the ultrastructural appearance of axons was normal in PLP/CFA-injected mice in comparison with CFA(control)-injected mice (Fig 7F1–4). Quantification of myelin sheath thickness (Fig 7I) and axon numbers in the optic nerves (Fig 7J) revealed no differences between PLP/CFA-injected mice in comparison with CFA-injected control mice (Fig 7I and J).

## Discussion

The starting point of our study was the finding that CASPR1 and CNTN1 are associated with synaptic ribbons in the retina. CASPR1/CNTN1 were previously characterized as a component of the paranodal region of myelinated axons (Rasband & Peles, 2015). Therefore, the co-immunoprecipitation of CASPR1/CNTN1 with immuno-isolated synaptic ribbons (Fig 1) was initially surprising because the retina contains virtually no myelinated axons and certainly no myelin in close vicinity to synaptic ribbons. Furthermore, CASPR1 was not found in synaptic ribbon proteome approaches from other groups (Uthaiah & Hudspeth, 2010; Kantardzhieva et al, 2012). Clearly, these latter two proteome studies used different antibodies and purification strategies than our present study.

The co-enrichment of CASPR1 with immunopurified synaptic ribbons was corroborated by immunolabeling analyses. Independent CASPR1 antibodies directed against different epitopes showed an enrichment of CASPR1 in the retinal synaptic layers (Appendix Figs S3, S6 and S10). These data confirm previous synaptic immunolocalization results in different regions (Sousa et al, 2009; O'Brien et al, 2010; Davisson et al, 2011; Lysakowski et al, 2012; Sedó-Cabezón et al, 2015). High-resolution confocal microscopy and super-resolution SIM analyses demonstrated enrichment of CASPR1 close to synaptic ribbons (Appendix Figs S5, S6B and C, and S7). At that location, CASPR1 largely co-localized with presynaptic active zone markers (RIMs, CASK) at the light microscopic level indicating a presynaptic localization of CASPR1 close to the ribbon (Appendix Figs S7 and S8). CNTN1 was also enriched close to the ribbons in retinal synapses (Fig 3A; Appendix Figs S5C and S10). Finally, immunogold electron microscopy confirmed enrichment of CASPR1 and CNTN1 at the synaptic ribbon complex in addition to a localization to pre- and postsynaptic plasma membrane in close vicinity to the synaptic ribbon (Fig EV1). Since CNTN1 forms a cis-complex with CASPR1 at the paranodal region (Rasband & Peles, 2015), these proteins might form a similar complex also at retinal ribbon synapses. This conclusion is supported by our data which show that CASPR1 and CNTN1 are both components of a RIBEYE-containing protein complex (Fig 1).

We demonstrate a rapid and strong auto-immune response against CASPR1 in the early pre-clinical phase of two different EAE mouse models (MOG and PLP mouse model of EAE/optic neuritis). This auto-immune response most likely is generated by epitope spreading. Epitope spreading is a common though incompletely understood phenomenon in auto-immune diseases including multiple sclerosis (McMahon et al, 2005; Flytzani et al, 2015). These auto-antibodies can reach synapses in the brain as judged by their presence in the CSF (Fig EV2A and F), i.e., beyond the blood barrier, and could be responsible for the synaptic decrease in the CASPR1/CNTN1 complex at retinal synapses (Figs 2, 3 and 7; Appendix Fig S13). The large extracellular portions of CASPR1/CNTN1 are exposed to the extracellular space and are thus accessible for binding to auto-antibodies. CASPR1/CNTN1 bound to ribbon-associated synaptic vesicles could be targeted to the ribbon via endocytosis that is known to be enriched at the peri-active zone (Wahl et al, 2013, 2016). In agreement with this proposal, we found CASPR1-positive vesicles in close vicinity to the peri-active zone of rod photoreceptor synapses (Fig EV1B3).

In humans, auto-antibodies against CASPR1 have recently been discovered as auto-antigens in patients with painful inflammatory neuropathies (Doppler et al, 2015). CASPR1-positive peripheral neuropathy has even been proposed to be a novel disease entity because the disease is more severe compared to CASPR1-negative patients and more difficult to treat (Doppler et al, 2016). However, CASPR1 as an auto-antigen targeted by epitope spreading in the non-myelinated retina during the induction phase of experimental optic neuritis was not the first thing one would have expected. Among the different clinical manifestations of human MS, optic neuritis represents one of the most frequent events and often occurs as the first sign of the disease. Within the spectrum of clinically isolated syndromes, it is a very homogenous disease condition with a predictable extent of neurodegeneration affecting retinal ganglion cells and optic nerve axons (Trip et al, 2005; Henderson et al, 2010). It has been shown that neurodegeneration in patients with a first episode of optic neuritis occurs rapidly within a few weeks with thinning of the retinal nerve fiber layer, the part of the retina which contains the non-myelinated optic nerve axons (Henderson et al, 2010). The retinal nerve fiber layer has also been shown to undergo thinning in eyes of MS patients who have not been clinically affected by optic neuritis (Henderson et al, 2008) and to reflect global neurodegenerative processes in MS as it correlates with brain atrophy (Gordon-Lipkin et al, 2007; Saidha et al, 2012). Additionally, it has been proposed that neuronal pathology affecting the inner and outer nuclear layer of the retina can develop independent from optic nerve changes (Green et al, 2010; Saidha et al, 2011).

Interestingly, reduction in the expression level of CASPR1/CNTN1 in pre-clinical optic neuritis/EAE is associated with a concomitant decrease in RIBEYE expression indicating a close functional connection of these proteins in retinal ribbon synapses (Figs 2B2, D2, 3B2, C and 4B2, and EV5). These findings suggest that synaptic ribbons are functionally linked to the CASPR1/CNTN1 network at retinal synapses and could thus be considered as an intracellular "effector" of the CASPR1/CNTN1 synaptic adhesion complex. The auto-antibodies might induce the observed decrease in CASPR1/CNTN1 and synaptic ribbon. The mechanisms how this signal is transduced to the ribbon is unclear and need to be further investigated. The presence of CASPR1/CNTN1 directly at the

presynaptic ribbon could indicate that these proteins are subject to endocytosis close to the active zone thus reaching the synaptic ribbon. Alternatively, CASPR1/CNTN1 at the base of the synaptic ribbon could transduce signals that finally could affect the stability of the synaptic ribbon. CASPR1 contains a protein 4.1 binding site (Horresh et al, 2010) that could transduce changes from the extracellular space to an intracellular re-organization of the presynaptic cytoskeleton. In line with this proposal, protein 4.1 is expressed in the presynaptic terminal (Sanuki et al, 2015).

Interestingly, the optical recording of synaptic activity in early, pre-clinical MOG/CFA mice revealed obvious similarities to synaptic defects observed in the RIBEYE knockout mouse (Maxeiner et al, 2016). Similar to retinal synapses of the RIBEYE knockout (analyzed by electrophysiology at the rod bipolar/amacrine AII synapse), retinal synapses in MOG/CFA mice (as analyzed by SypHy activity) also showed a reduction in both fast and slow sustained exocytosis further pointing to the possibility that CASPR1 antibodies affect the synaptic ribbon complex which in turn could be responsible for the altered synaptic vesicle cycling in rod photoreceptor ribbon synapses of EAE mice. The optical recording of synaptic activity with the SypHy mice (Fig 6) and the recording of the optokinetic responses in the visual behavior experiments (Fig 4F) demonstrated that retinal synapses are compromised but clearly are still present/ alive and functionally active. This is an important issue considering the strong recruitment of the complement system to retinal synapses in early EAE (Figs EV3 and EV4). Some basal expression of complement proteins at retinal synapses under control conditions could reflect its recently discovered role in normal synaptic physiology (Yuzaki, 2017). The strong activation of the complement system at retinal synapses in early, pre-clinical EAE likely exerts a negative impact on retinal synapse function (Stephan et al, 2012).

Importantly, the synaptic CASPR1/CNTN1 complex is strongly decreased and the functional synaptic changes occur at an early, pre-clinical stage of optic neuritis/EAE before the onset of obvious morphological alterations in the optic nerve that subsequently occur at a later time (Fairless et al, 2012; Stojic et al, 2018). This was shown by various morphological analyses (Figs 5 and 7; Appendix Fig S12), including immunolabeling analyses of the paranodal regions of the optic nerve that are an early and sensitive readout for functional changes in the optic nerve (Stojic et al, 2018). Thus, the synaptic changes in the retina precede the morphological changes in the optic nerve that subsequently develop during optic neuritis/MS. These findings suggest that the synaptic changes are not secondary to alterations in the optic nerve but could contribute to ensuing optic nerve pathology/demyelination. In support of this assumption, synaptic dysfunctions have been proposed to lead to subsequent excitotoxic cell death and loss of axons, e.g., in hippocampus, striatum, and cerebellum (Mandolesi et al, 2015; Calabrese et al, 2015; Stampanoni Bassi et al, 2017). We cannot exclude though that remote functional changes in the optic nerve preceding demyelination and possibly escaping morphological alterations could influence intraretinal synaptic pathology.

In conclusion, our data reveal the novel finding that the autoimmune response against CASPR1, a known antigen in autoimmune disease (Querol & Illa, 2015; Stathopoulos et al, 2015), not only targets the paranodal region of myelinated nerves, but also targets retinal synapses both at a structural and functional level at a very early time point. Therefore, the pathophysiological role of an auto-reactive immune response against CASPR1 is not restricted to the auto-immune targeting of these proteins at the node of Ranvier, as previously assumed (Wolswijk & Balesar, 2003; Coman et al, 2006; Querol & Illa, 2015; Stathopoulos et al, 2015) but also compromises retinal synapses, that also express CASPR1 and CNTN1. This targeting of retinal synapses is manifested at an early time point before morphological alterations of the optic nerve are visible. So far, retinal synapses were not considered to be relevant for neuroinflammatory changes in MS. Our findings show that this view is not justified and raise the possibility that retinal synaptic alterations could represent one of the earliest pathophysiological events in the development of EAE/MS.

# Materials and Methods

## Animals

Experiments were performed on tissues obtained from C57BL/6 mice of the indicated sex and age. Animal care and all experimental procedures that involved mice were performed according to the guidelines of the German Animal Protection Law (Tierschutzgesetz) and were reviewed and approved by the animal welfare and ethics committee of the Saarland University and the local authorities. Mice were kept under standard light/dark cycle and supported with standard food and water *ad libitum*. For the mass spectrometry analyses, bovine eyes were used from a local slaughterhouse as reviewed and approved by the local authorities. Retinas were isolated from the eyes as previously described (Schmitz et al, 2000). RIBEYE knockout mice were previously described (Maxeiner et al, 2016) and used as described in the respective experiments.

## Primary antibodies

- Anti-RIBEYE(B). Rabbit polyclonal antiserum against RIBEYE(B)-domain (U2656; Schmitz et al, 2000) was used in a 1:1,000 dilution for immunofluorescence microscopy.
- Anti-RIBEYE(B). A mouse monoclonal antibody (clone 2D9) was raised against the carboxyterminal 12 amino acids (KHGDNREHPNEQ) of mouse RIBEYE which is extremely conserved between species (100% identity at the amino acid level between mouse, bovine, and human RIBEYE(B)-domain/CtBP2). The antibody supernatant was used for Western blots in a 1:400 dilution and for immunofluorescence microscopy in a 1:300 dilution of an antibody stock solution that had an immunoglobulin concentration of 0.8 mg/ml.
- Anti-CASPR1 (rabbit polyclonal) was purchased from Abcam (#ab34151). This antibody is directed against the intracellular carboxyterminus of mouse CASPR1 (aa1,350–aa1,385) and was used in a 1:1,000 dilution for Western blots, in a 1:100 dilution for immunofluorescence microscopy on the retina, and in a 1:500 dilution for immunolabeling of the optic nerve. The corresponding peptide used for immunization was purchased from Abcam (ab34150) for pre-absorption experiments (Appendix Figs S2 and S4).
- Anti-CASPR1 (mouse monoclonal antibody; clone30-p190) generated against the extracellular aminoterminus of CASPR1, i.e., aa1–1,513. This monoclonal antibody was purchased from "antikörper-online" (ABIN967781) and used in a 1:1,000 dilution for

Western blotting and in a 1:100 dilution for immunofluorescence microscopy.

- Anti-CASPR1. A mouse monoclonal antibody (clone 5F9) was raised against an internal peptide sequence (31mer) of CASPR1 located in the discoidin domain (SGAWGWGYYGCNEELVGPLYARSLGAS-SYYG) of mouse CASPR1. This sequence stretch is highly conserved between species (≈94% identity at the amino acid level between mouse, bovine, and human CASPR1). The antibody supernatant was used for Western blots in a 1:100 dilution and for immunofluorescence microscopy in a 1:50 dilution of an antibody stock solution that had an immunoglobulin concentration of ≈0.5 mg/ml.
- Anti-contactin1 (mouse monoclonal antibody clone S73-2D) obtained from Sigma-Aldrich (Order number: SAB5200075) used at a 1:300 dilution in Western blotting and used in a 1:50 dilution for immunofluorescence microscopy.
- Anti-contactin1. Rabbit polyclonal antibody (Abcam; #ab66265), raised against a synthetic peptide derived from residues 250–350 of mouse contactin1. This antibody was used in a 1:500 dilution for immunofluorescence microscopy, for Western blotting in a 1:500 dilution.
- Anti-complement C3 protein A-purified rabbit polyclonal antibody against the carboxyterminal region (aa1,600-to the C-terminus of human complement C3 protein, obtained from Abcam, #EPR 19394). This antibody was used in a 1:500 dilution for immunofluorescence microscopy, for Western blots in a 1:2,000 dilution.
- Anti-complement C5b-9 protein A-purified mouse monoclonal antibody (IgG2a; clone number aE11) raised against the neo-epitope exposed on complement protein 9 (C9) when incorporated into the terminal complement complex (TCC; Fluiter et al, 2014; Michailidou et al, 2015). This antibody was obtained from Abcam (#ab66768; clone αE11) and used for immunofluorescence microscopy in a 1:50 dilution.

Additional primary antibodies used in the study: anti-visual arrestin [Santa Cruz (E12, sc-34547)] affinity-purified goat polyclonal antibody against a peptide from an internal region of human visual arrestin, used in a 1:250 dilution for immunofluorescence microscopy; anti-mGluR6; rabbit polyclonal antiserum (Katiyar et al, 2015) used in a 1:200 dilution for immunofluorescence microscopy; anti-Cask mouse monoclonal antibody (Anjum et al, 2014), used in a 1:200 dilution for immunofluorescence microscopy; anti-RIM1/2 (Anjum et al, 2014); rabbit polyclonal antiserum used in a 1:500 dilution for immunofluorescence microscopy; anti-dynamin1xb (clone 9E10; Eich et al, 2017), mouse monoclonal antibody used at a 1:300 dilution for Western blot and immunofluorescence microscopy; anti-PSD95 (NeuroMABs) used in a 1:200 dilution for immunofluorescence microscopy; anti-synaptophysin (SIGMA, clone SVP-38; #SAB4200544) mouse monoclonal antibody against synaptophysin, purified immunoglobulin, used at a 1:1,000 dilution for Western blots; anti-mCherry, mouse monoclonal antibody (Abcam 1C51, ab125096) used at 1:2,000 dilution for Western blot analyses; Anti-Na$_v$-channel pan mouse monoclonal antibody (Sigma, clone K58/35; S8809) used at dilution of 1:50 for immunolabeling.

## Secondary antibodies

Secondary antibodies for immunofluorescence microscopy: Chicken anti-mouse immunoglobulins conjugated to Alexa488

(Invitrogen, Carlsbad, CA, USA, #A21200). Donkey anti-mouse immunoglobulins conjugated to Alexa568 (Invitrogen, #A10037). Donkey anti-rabbit immunoglobulins conjugated to Alexa568 (Invitrogen, #A10042). Chicken anti-rabbit immunoglobulins conjugated to Alexa488 (Invitrogen, #A21441). Goat anti-mouse immunoglobulins conjugated to Alexa647 (Invitrogen, #A21236). Monovalent Fab fragments rabbit anti-mouse [unconjugated; Fab rabbit anti-mouse IgG (H&L); Rockland Immunochemicals, #810-4102 via Biomol GmbH, Hamburg, Germany], used for immunofluorescence microscopy in a 1:50 dilution. Chicken anti-goat immunoglobulins conjugated to Alexa594 (Invitrogen, #A21468). Secondary antibodies for Western blot analyses: goat anti-rabbit immunoglobulins conjugated to peroxidase (Sigma, #A6154), used in a 1:5,000 dilution for Western blot analyses; goat anti-mouse immunoglobulins conjugated to peroxidase (Sigma, #A3673), used in a 1:5,000 dilution for Western blot analyses.

## Plasmids

pCASPR1-mCherryN1. CASPR1 full-length cDNA was cloned into pmCherryN1 vector (Dembla et al, 2014). For this purpose, full-length CASPR1 cDNA was amplified via PCR using forward primer AAAAGAATTCTGGCCACCATGATGAGT CTCCGGC, reverse primer AAAAACCGGTGGTTCAGACCTGGACTCCTCC and CASPR1 cDNA (BC156962.1; IMAGE clone:#100063633) as template. The full-length PCR product was cloned into the EcoRI/AgeI sites of pmCherryN1 and verified by sequencing.

## Immunoprecipitation of synaptic ribbons from the bovine retina

For immunoprecipitation experiments, isolated bovine retinas were incubated in 1 ml lysis buffer, containing 100 mM Tris–HCl, pH 8.0, 150 mM NaCl, 1 mM EDTA, and 1% TX-100 for 45 min on vertical rotator at 4°C. The sample was mechanically cracked by forcefully ejecting the retinal lysate through a 20G needle on ice. This procedure was repeated 20 times. The sample was sonicated at 10% output for 20 cycles on ice. After lysis, the extract was centrifuged at 16,060 g for 30 min in an Eppendorf centrifuge at 4°C. The lysate was collected in a new Eppendorf tube and above mentioned step was repeated one more time to remove all cell debris. The lysate was pre-cleared by the addition of 15 μl of normal mouse IgG (Santa Cruz, sc-2025) and 20 μl of washed protein A-Sepharose beads (2-h incubation at 4°C with an overhead rotator). Next, samples were centrifuged at 13,000 rpm for 30 min at 4°C. The supernatant was split into two equal volumes, one for the control and one for the experimental assay. For the negative control, 20 μl of normal mouse IgG was added to the cleared lysate; for the experimental assay, 20 μl of mouse monoclonal antibody against RIBEYE (clone 2D9) was added to the pre-cleared lysate. Samples were incubated overnight at 4°C on an overhead rotator. Next, beads were allowed to settle down by gravity on ice (for ≈20 min). Supernatant was collected in different tubes, and beads were resuspended in 1.0 ml of lysis buffer and washed thrice by repeated centrifugation (855 g, 1 min, 4°C). The final pellet was boiled in 10 μl sample buffer, subjected to SDS–PAGE, and probed by Western blot analyses with the indicated antibodies.

## Mass spectrometric analyses

### Gel electrophoresis of proteins and mass spectrometry

Eluted proteins were separated on 4–12% gradient gels (NuPAGE®, ThermoFisher Scientific) and prepared for mass spectrometry as described before (Fecher-Trost *et al*, 2013). For tryptic in-gel digestion, the gel pieces were incubated with 15–20 μl of porcine trypsin (20 ng/μl, Promega) at 37°C overnight. Resulting peptides were extracted twice by shaking the gel pieces in aqueous extraction buffer (2.5% formic acid, 50% acetonitrile). Extracted peptides were concentrated in a vacuum centrifuge and resuspended in 20 μl of 0.1% formic acid.

### Nano-LC-HR-MS/MS

One third of the tryptic peptides of each antibody purification was analyzed by high-resolution nanoflow MS/MS (LTQ Orbitrap Velos Pro coupled to Ultimate 3000 RSLC nano system equipped with an Ultimate3000 RS autosampler, ThermoFisher Scientific, Dreieich, Germany). Peptides were trapped on a trap column (C18, 75 μm × 2 cm, Acclaim PepMap100C18, 3 μm, ThermoFisher) and separated on a reversed phase column (nano viper Acclaim PepMap capillary column, C18; 2 μm; 75 μm × 50 cm, ThermoFisher) at a flow rate of 200 nl/min. The gradient was build with buffer A (water and 0.1% formic acid) and B (90% acetonitrile and 0.1% formic acid) using a gradient (4–55% buffer B in 56 min; 55–90% buffer B in 7 min). The effluent was directly sprayed into the mass spectrometer through a coated silica emitter (PicoTipEmitter, 30 μm, New Objective, Woburn, MA, USA) and ionized at 2.2 kV. MS spectra were acquired in a data-dependent mode. For the collision-induced dissociation (CID) MS/MS top 10 method, full-scan MS spectra (m/z 300–1,700) were acquired in the Orbitrap analyzer using a target value of $10^6$. The 10 most intense peptide ions (charge states ≥ 2) were fragmented in the high-pressure linear ion trap by low-energy CID with normalized collision energy of 35%.

### LC-MS data analysis

Fragmented peptides masses were analyzed by using the MASCOT algorithm and Proteome Discoverer 1.4 software (Thermo Fisher). For this purpose, peptides were matched to tandem mass spectra by Mascot version 2.4.0 by searching an in house modified SwissProt database (basis version 2012_04, number of protein sequences: 535.255, taxonomy mammalia: 65.787). $MS^2$ spectra were matched with a mass tolerance of 7 ppm for precursor masses and 0.5 Da for fragment ions. Tryptic digest with up to two missed cleavage sites was allowed. Deamidation of asparagine and glutamine, acetylation of lysine, and oxidation of methionine were set as variable modifications, and cysteine carbamidomethylation was set as a fixed modification.

## Generation of transgenic SypHy mice that express SypHy under the control of the rod opsin promotor

A transgenic mouse was generated in which the pH-sensitive SypHy (acronym for a fusion construct of pH-sensitive GFP fused to the lumenal domain of the synaptic vesicle protein synaptophysin; Linares-Clemente *et al*, 2015) was expressed under the control of the mouse rod opsin promotor (Geppert *et al*, 1994). The mouse opsin promotor was cloned into the HindIII/SalI site,

the SypHy cDNA into the XhoI/BamHI site, and the human growth hormone polyadenylation signal into the BamHI/NotI site of a pEGFP-N1-based plasmid vector. For pronucleus injection, the insert was excised via HindII/NotI and gel-purified using standard methods.

Pronucleus injection was performed at the IBF facility (University of Heidelberg, Germany) by Frank Zimmermann/Sascha Dlugosz. Presence of the transgene in founder mice was verified by PCR using the primer pair F2 (CCACGGAGATCCGCCGAGCA)/R2 (CGCCCTC GGATGTGCACTTGA). Transgenic animals were back-crossed into C57BL/6 background for more than 20 generations.

## Induction of Experimental Auto-immune Encephalomyelitis (EAE) in female C57BL/6 mice

Experimental auto-immune encephalomyelitis (EAE), a frequently used mouse model of multiple sclerosis (Robinson *et al*, 2014), was induced in female C57BL/6 mice largely as previously described (Williams *et al*, 2011). For EAE induction, female C57BL/6 mice older than 10 weeks (body weight between 20 and 25 g) were selected. Mice were injected subcutaneously into the axilla and groin with a total of 200 μg of the encephalitogenic $MOG_{35-55}$ peptide of mouse myelin oligodendrocyte glycoprotein (MEVG-WYRSPFSRVVHLYRNGK; > 90% purity, synthesized by Dr. Martin Jung, Institute of Medical Biochemistry and Molecular Biology, Medical School Homburg, Saarland University) in sterile water (2 mg/ml) emulsified with an equal volume of complete Freund adjuvant (CFA), consisting of incomplete Freund adjuvant (iCFA, Sigma; #F5506) to which 10 mg/ml inactivated *Mycobacterium tuberculosis* were added (Fisher Scientific #10218823), as previously described (Williams *et al*, 2011). 200 ng of pertussis toxin (PTX) from *Bordetella pertussis* (List Biological Laboratories Inc. #181, via Biotrend, Cologne, Germany) in a volume of 100 μl sterile $H_2O$ was injected intraperitoneally on the same day (day 0, 1–2 h after $MOG_{35-55}$ peptide injection) and also on the subsequent day (day 1, 16–20 h after first PTX injection). Controls were injected with CFA only, i.e., without $MOG_{35-55}$ peptide. All other treatments for the control injections, e.g., pertussis toxin injection, were done identically as described for MOG/CFA injection. Subsequent analyses were done blindly; i.e., the experimenter was not aware whether a mouse was MOG/CFA-injected or CFA (control)-injected. To exclude effects which depend on the preparation of the MOG/CFA suspension (emulsion), we used two batches, one home-made and a second one obtained as part of a kit from Hooke Laboratories with pre-made, ready-to-go suspensions ($MOG_{35-55}$/CFA Emulsion PTX, Hooke Laboratories, Lawrence. MA, USA #EK-2110 and CFA control kit # CK-2110). This was done to exclude effects that might have arisen due to any contaminant in the self-mixed reagents.

For PLP immunization, female C57BL/6 mice (8 weeks of age) were injected with 200 μg of the proteolipid protein peptide $PLP_{180-199}$ (Terry *et al*, 2016). $PLP_{180-199}$ (WTTCQSIAFPSKTSASIGSL; synthesized by Dr. Rudolf Volkmer, Charité-University Medicine Berlin) was dissolved in PBS, emulsified in an equal volume of CFA, and injected subcutaneously into four locations in the flanks. Immediately afterward, and again 48 h later, mice received intraperitoneal injections of 200 ng of pertussis toxin. As for MOG EAE, CFA-immunized mice were treated identically, but with emulsion lacking the $PLP_{180-199}$ peptide.

## Embedding of tissue for immunofluorescence microscopy on semi-thin sections

Tissue embedding was done exactly as previously described (Wahl *et al*, 2016; Eich *et al*, 2017). After deep isoflurane anesthesia, mice of both sexes with an age older than 10 weeks were euthanized by cervical dislocation. For the EAE experiments, only female animals were used. Eyes were isolated within 5 min post-mortem and punctured with a 20G needle at the equatorial region of the eye. The anterior part of eye (including lens and vitreous body) was removed by a circular cut in the equatorial plane of the eye. The posterior eyecups with the attached retinas were plunge-frozen in liquid nitrogen-cooled isopentane as previously described (Schmitz *et al*, 2000). Frozen tissue was freeze-dried under liquid nitrogen for 2 days, equilibrated to room temperature, and infiltrated with Epon overnight at 28°C in a rotator (2 rpm). Infiltrated tissue was polymerized for ≈24 h at 60°C. For Appendix Fig S3A, we used (non-embedded) 10-μm-thick cryostat sections that were prepared from the mouse retina as previously described (Schmitz *et al*, 2000) and collected on uncoated glass slides (SuperFrost, Menzel, Germany).

## Immunolabeling of semi-thin sections for immunofluorescence microscopy

Immunolabeling was performed on 0.5-μm-thin ("semi-thin") sections after resin removal exactly as previously described (Wahl *et al*, 2016; Eich *et al*, 2017). Semi-thin sections were collected on glass coverslips, and the resin was removed before the immunolabeling procedure, as described (Wahl *et al*, 2016; Eich *et al*, 2017). From the immunolabeled sections, images were acquired with a Nikon A1R confocal microscope. In the described double-immunolabeling analyses, the two primary antibodies were always generated in different animal species (i.e., mouse and rabbit, respectively) and therefore could be applied simultaneously. After several washes with PBS to remove unbound primary antibody, binding of the primary antibodies was detected by incubation with the indicated secondary antibodies conjugated to the respective fluorescent dye (1:1,000 dilution; 1 h, room temperature, 21°C). Controls were done by omitting the primary antibodies and using the secondary antibodies only or by using irrelevant primary antibodies. For controls in double-immunolabeling experiments, one (of the two) primary antibodies was omitted to judge on the specificity of the immunosignals, i.e., to check for possible cross-talks between the two different immunosignals. Sections were analyzed on a Nikon A1R confocal microscope, as previously described (Wahl *et al*, 2013, 2016; Eich *et al*, 2017). Super-resolution structured illumination microscopy (SR-SIM) was performed with an Elyra PS1 setup (Zeiss) equipped with ZEN software, exactly as previously described using a 63× Plan Apo objective (N.A. 1.4).

## Triple immunolabeling of semi-thin sections

Triple-immunolabeling analyses with three different primary antibodies (with two antibodies from an identical species; i.e., mouse or rabbit) were performed, as previously described (Eich *et al*, 2017). In brief, two of the three primary antibodies that were generated in different species (i.e., a mouse primary antibody and a rabbit primary antibody) were incubated simultaneously overnight at 4°C

at the indicated dilutions. After incubation in the primary antibody solutions, semi-thin sections were washed thrice with PBS to remove unbound primary antibodies and next incubated with the fluorophore-conjugated secondary antibody (1:1,000 dilution; 1 h, 21°C). Depending on the species in which the third primary antibody was generated, sections were next pre-incubated with either anti-rabbit or anti-mouse polyclonal monovalent Fab fragments and incubated for 3–4 h at room temperature. If the third primary antibody was from mouse, residual binding sites of tissue-bound mouse primary antibody were blocked using rabbit polyclonal, monovalent anti-mouse IgG Fab fragments (Rockland Immunochemicals #810-4102; 1:50 dilution from a 1.0 mg/ml stock; 3 h, 21°C). If the third primary antibody was from rabbit, residual binding sites of tissue-bound rabbit primary antibody were blocked using goat polyclonal, monovalent anti-rabbit IgG Fab fragments (Rockland Immunochemicals #811-1102 from Biomol GmbH; 1:50 dilution from a 1.0 mg/ml stock; 3 h, 21°C). After three washes with PBS, the third primary antibody was added and incubated overnight (4°C) at the indicated dilution. Binding of the third primary antibody was detected with donkey anti-rabbit/donkey anti-mouse secondary antibodies conjugated to Alexa647, as indicated in the respective experiments. Sections were washed with PBS for three times and mounted on glass slides with N-propyl gallate in glycerol, as described (Wahl *et al*, 2016; Eich *et al*, 2017). Controls were done by performing the described immunolabeling procedure but with one (of the two) primary antibodies generated in the same species omitted to judge on the specificity of the immunosignals and to check for possible cross-talks between the two different immunosignals obtained with the primary antibodies generated in the same species. No cross-talk signal was observed in these control incubations.

## Immunolabeling of cryostat sections

Immunolabeling of cryostat sections was performed as previously described (Schmitz *et al*, 2000; Alpadi *et al*, 2008).

## Quantification of immunofluorescence signals

For quantitative analysis, images were acquired using the NIS Elements software (NIS Elements AR 3.2, 64 bit) of the A1R confocal microscope (Nikon). At first, images were acquired from immunolabeled sections that were obtained from retinas of either CFA-injected control mice or MOG/CFA-injected mice. The analyses were done blindly; i.e., the experimenter was not aware whether sections were from MOG/CFA-injected or control-injected mice. Identical conditions were maintained for all sections (from MOG/CFA-injected and CFA-injected mice) using the "re-use" settings option of the NIS elements software, as previously described (Wahl *et al*, 2016; Eich *et al*, 2017). For quantification, images were analyzed using Fiji ImageJ 1.5 h software (NIH) and the fluorescence intensity was determined as integrated density. In all fluorescence quantifications, values were normalized and CFA values were set to 100% if not denoted otherwise. Values from MOG/CFA sections were normalized to the CFA reference. All the analyses were performed without changing any parameters in the individual channels. The areas of the outer plexiform layer (OPL) and inner plexiform layer (IPL) were selected by means of the RIBEYE immunosignals. RIBEYE is a synaptic component in both the OPL

and IPL (Schmitz *et al*, 2000) and thus provided a reference for the identification of the synaptic layers in the retina. The integrated density was measured for the identified synaptic areas. Then, the identical region of interest (ROI) was used to measure the integrated density of either CASPR1, CNTN1, v-Arrestin, complement protein C3, or complement C5b-9 as indicated in the respective experiments (Figs 2, 3, EV3 and EV4, Appendix Fig S11). In case of C5b-9 immunolabeling analyses, MOG/CFA values were normalized to 100% due to the absence of C5b-9 immunosignals in the synaptic layers of CFA-injected mice (Appendix Fig EV4). Statistical analyses were performed with Wilcoxon–Mann–Whitney *U* rank test, as described below. Results of the statistical analyses for the respective experiments are displayed either directly in the figure or in the figure legend.

### Collection of blood from CFA and MOG/CFA mice

Blood collection before injection was done from the tail veins of the respective mice. For blood collection from CFA- and MOG/CFA-injected mice, animals were euthanized by cervical dislocation after isoflurane anesthesia and blood was collected by puncturing the left ventricle of the heart. Sodium citrate was added to all blood samples to prevent clotting of the blood.

### Collection of cerebrospinal fluid (CSF)

Cerebrospinal fluid was collected from the cerebellomedullary cistern of the fourth ventricle (cisterna magna) using a Hamilton syringe (Liu & Duff, 2008).

### ELISA testing of mouse blood samples for reactivity against retinal proteins

In brief, mouse retinas were incubated in 0.2 ml of lysis buffer, containing 100 mM Tris–HCl, pH 8.0, 150 mM NaCl, 1 mM EDTA, and 0.1% (v/v) Triton X-100 for 15 min on vertical rotator at 4°C. Samples were mechanically cracked by forceful ejection through a 20G needle and subsequent sonication on ice (Bandelin; Sonoplus; 1% output for 20 half-second pulse ON/OFF cycles). Afterward, the extracts were centrifuged twice at 13,000 rpm (30 min, 4°C) to remove all cell debris. The resulting supernatants were used for coating of the ELISA plates. Protein concentration of the extracts was determined as previously described (Eich *et al*, 2017). For coating, 96-well microtiter plates were used (M4561-40EA; Sigma). Retinal protein lysate was diluted to 50 μg/ml concentration using carbonate buffer (36 mM $Na_2CO_3$ and 57 mM $NaHCO_3$, pH 9.6). 100 μl of diluted retinal lysate was added to each well, and plate was coated overnight at 4°C. For blocking, the plate was washed two times with PBS and to each well 200 μl of 4% skim milk solution in PBS was added. The plates were incubated for 2 h at 21°C. Again, washing step was repeated twice. Serum samples from MOG/CFA and CFA control mice obtained either directly before and 7–9 days after injection with MOG/CFA or CFA were diluted to 1:100 in 1% skim milk solution in PBS (90 min, 21°C). Mouse monoclonal primary antibodies against RIBEYE (clone 2D9) and SV2 served as a positive controls for the immunodetection procedure. Secondary antibodies only (without primary antibody) served as negative control. After incubation with the respective serum

samples (from MOG/CFA or CFA mice) or the primary positive control antibodies and the indicated negative controls, the 96-well plates were washed with 300 μl of 1× PBS, 0.05% Tween-20 per well for 4–5 times. Secondary antibody, i.e., goat anti-mouse IgG, IgA, IgM immunoglobulins conjugated to peroxidase (Sigma, #SAB3701048), was diluted 1:2,000 in 1% skim milk solution in PBS and 100 μl of the secondary antibody dilution was added to each well and incubated at 21°C for 90 min. Again, the plate was washed 4–5 times with 1× PBS, 0.05% Tween-20. 100 μl of ready-to-go TMB color substrate solution (T0440-100 ml; Sigma) was added to each well and incubated for 15 min at 21°C. After 15 min of development of the reaction product, absorbance was immediately measured at 655 nm using Bio-Rad iMARK™ microplate reader. For analysis, values from the pre-injection blood samples and negative controls (secondary antibody only) were pooled and absorbance from all the samples from three individual experiments were normalized to pre-injection control values and plotted as shown in the Fig 4A.

### Pre-embedding immunogold electron microscopy

Pre-embedding immunogold electron microscopy was performed, largely as previously described (Schmitz *et al*, 1996; Suiwal *et al*, 2017). Mouse retinas were flash-frozen in liquid nitrogen—cooled isopentane, as described (Schmitz *et al*, 2000; Alpadi *et al*, 2008), and 20-μm-thick cryostat sections were obtained using standard techniques. Cryostat sections were collected on glass coverslides that were freshly coated with a thick layer of gelatine (Suiwal *et al*, 2017). The thick layer of gelatine is important for later-on removal of the embedded tissue sections from the glass surface. Before incubation with the primary antibodies, sections were treated with 0.5% BSA in PBS (15 min, RT) to block unspecific binding sites. Binding of the primary antibodies (monoclonal anti-Caspr1 [5F9] and monoclonal anti-RIBEYE [2D9]; diluted in a 1:100 dilution each in blocking buffer) were performed for 3 h at RT. Incubation with PBS alone (with no primary antibody) served as negative control. After several washes with PBS to remove unbound antibody, sections were gently fixed with 2% paraformaldehyde in PBS for 5 min at RT. Following treatment with 0.5% BSA in PBS, sections were incubated with goat anti-mouse secondary antibodies conjugated to ultrasmall (1 nm diameter gold particles) gold particles (1:100 dilution in blocking buffer; 1 h, RT). Unbound secondary antibody was removed by several washes with PBS, and sections were next fixed with 2.5% glutaraldehyde (15 min RT). Then, ultrasmall gold particles were silver-enhanced for 20 min using a commercial silver enhancement kit (HQ Silver Enhancement Kit; Nanoprobes) in darkness according to the manufacturer's instruction. After silver enhancement, sections were osmicated with 2% $OsO_4$ (15 min, RT), treated with 2% uranyl acetate (15 min, RT, incubation in the dark), and dehydrated using an ascending concentration series of ethanol. After a 5-min incubation in propylene oxide, sections were infiltrated with Epon. The sections were polymerized into an Epon block as previously described (Suiwal *et al*, 2017).

### Post-embedding immunogold electron microscopy

Post-embedding immunogold electron microscopy was performed exactly as previously described (Schmitz *et al*, 2000). The

monoclonal anti-CNTN1 antibody was used in a 1:50 dilution; monoclonal anti-RIBEYE in a 1:500 dilution; the secondary goat-anti-mouse conjugated to 5 nm gold particles was used in a 1:500 dilution.

## Quantification of nodal and paranodal length in the optic nerve

For quantification of nodal and paranodal length, 10-μm-thick horizontal cryostat sections of the optic nerves were collected on glass slides. Sections were heat fixed for 10 min followed by permeabilization for 20 min with 0.1% TX-100 and 0.5% BSA in PBS (pH 7.5) at RT. Sections were labeled with anti-Nav-channel antibody (mouse monoclonal antibody, dilution 1:50) to mark the nodal region and anti-CASPR1 antibody (rabbit polyclonal antibody, dilution 1:500) to visualize the paranodal regions. Images were taken with a 60× objective (Plan Apo λ; NA 1.40; WD 0.13) using a A1R confocal microscope (Nikon). Identical conditions were maintained for all sections (from MOG/CFA-injected and CFA-injected mice) using the "re-use" settings option of the NIS elements software, as previously described (Wahl *et al*, 2016; Eich *et al*, 2017). Using the NIS elements software nodal and paranodal length was determined by manually marking the endpoints of each immunofluorescent signal. Individual length was than calculated by the software as distance between these endpoints. For further analysis, data obtained by the NIS software were exported to Excel and plotted as average length (μm) for each condition (self-made emulsion; commercial emulsion; PLP emulsion).

## Quantification of myelin thickness of the optic nerves

For the analysis of myelin sheath thickness of axons in the optic nerve, isolated optic nerves were first processed for standard electron microscopy as previously described (Maxeiner *et al*, 2016). Ultrathin (≈70-nm-thick) cross-sections of optic nerves were prepared and imaged using a Tecnai Biotwin electron microscope at 16,500× magnification. For analysis, images were exported as tiff files including metadata and scaling information. Myelin sheath thickness was determined using NIS elements software similarly as described above for the quantification of nodal and paranodal length. In brief, two endpoints (one at the outer rim of the myelin sheet and one at the inner border of the myelin sheet) were marked manually at individual axons and the distance in between these points was calculated in nanometers (using the scaling information of the images). Data were exported to excel and plotted as average thickness (in nanometers, nm).

## Quantification of optic nerve fiber numbers

Nerve fiber number was analyzed using the same specimens as used for quantification of myelin thickness. In order to observe a bigger area and not to miss subtle changes in fiber, number counting was performed on Richardson Blue-stained semi-thin sections. Images were taken using a A1R confocal microscope (Nikon) and either a 10× objective (Plan Fluor; NA 0.30; DIC) to cover the whole optic nerve cross-sectional area or a 60× objective (Plan Apo λ; NA 1.40; WD 0.13) to get a higher resolution particularly for counting thin axons that are not resolved by the 10× objective. Images were acquired by sub-dividing the cross-sectional area into four identical regions of interest. Image analysis was done with ImageJ. Using the "find maxima" plug-in with setting tolerance to 0% allowed an automated unbiased detection of axon number. Axon number of the four regions of interest were added to get the total fiber number per field of vision (60× objective). Total optic nerve fiber count was extrapolated, and data were plotted as total axon number/optic nerve for all conditions (home-made MOG suspension; commercial MOG suspension; PLP suspension).

## Quantification of immunofluorescent puncta in the OPL and IPL

Immunofluorescence images were acquired as described above und used for obtaining absolute puncta number in the respective area of interest. By using the ImageJ plug-in for the finding of signal maxima, puncta were calculated by using 50% tolerance for CASPR1, CNTN1, C3, and C5b-9 signals from OPL or IPL. For quantification of OPL puncta, puncta were measuring in immunolabeled retinal cross-section along a length of 65 μm. Puncta in the IPL were determined in a reference area of 11,160 μm$^2$. Values were exported to Excel and plotted in Appendix Fig S15.

## Preparation of retinal slices from MOG/CFA-injected transgenic SypHy reporter mice for the analysis of vesicle cycling in rod photoreceptor synapses

MOG/CFA- and CFA-injected SypHy reporter mice were sacrificed on day 7, day 8, or day 9 after injection and their retinas were isolated within 5 min of post-mortem as described in Wahl *et al* (2016). Briefly, enucleated eyes were punctured at anterior side using thin syringe needle and bisected at the equatorial plane. The posterior eye cup was transferred into low calcium solution (abbreviated as "LCS" in the text; containing 132 mM NaCl, 3 mM KCl, 1 mM MgCl$_2$ × 6H$_2$O, 0.5 mM CaCl$_2$, 10 mM sodium pyruvate, 10 mM glucose, 10 mM HEPES, pH 7.4; ≈300 mOsm/l). LCS was saturated with 5% CO$_2$, 95% O$_2$ before use. From the posterior eye cup, the retina was gently peeled off from the pigment epithelium. Four cuts were made in the retina so that it could be flat-mounted onto black-gridded nitrocellulose filter membranes (Millipore, #HABG01300) with ganglion cell side facing nitrocellulose membrane. Membrane filters with attached retina and some LCS (low calcium solution, to prevent drying of the retinas) were transferred to a silica sieve funnel, and gentle suction was applied to enhance adhesion of the retina to filter membrane via the attached syringe for strengthen the attachment of the retina to the filter.

The nitrocellulose filter with the attached retina on top was transferred to a glass slide, and some streaks of Vaseline were applied to the glass slide to prevent lateral movements of the filter during subsequent slicing. The glass slide with the retina attached on nitrocellulose filter was transferred to the cutting stage of Werblin-type tissue slicer. Retina slices of ≈200 μm thickness were sectioned with the slicer. Slices were then immediately transferred onto a glass coverslip with parallel streaks of Vaseline on it. The gaps between the streaks of Vaseline were filled with resting solution (RS; containing 132 mM NaCl, 3 mM KCl, 1 mM MgCl$_2$ × 6H$_2$O, 2 mM CaCl$_2$, 10 mM sodium pyruvate, 10 mM glucose, 10 mM HEPES, pH 7.4; ≈300 mOsm/l). RS was saturated with 5% CO$_2$, 95% O$_2$ before use. Slices were carefully picked from the cutting platform with fine tweezers, turned vertically so that all

retinal layers were visible from the photoreceptors on the free outer side of the slice to the ganglion cells that were facing toward the filter surface, and fixed between the Vaseline streaks.

### Optical recording of synaptic activity in MOG/CFA-injected SypHy reporter mice

All experiments were performed using 10- to 12-week-old SypHy transgenic reporter mice with C57BL/6 genetic background. Retinal slices ($\approx$200-$\mu$m-thick), prepared from MOG/CFA-injected transgenic SypHy reporter mice on day 9 after injection were incubated in the dark at 29°C temperature for 15 min in low calcium solution (LCS). The coverslip containing the retina slice was placed in the holding chamber and rinsed three times with 2 mM $Ca^{2+}$-containing resting solution (abbreviated as "RS" in the text) gassed with 5% $CO_2$, 95% $O_2$. After washing, the holding chamber was filled with RS to submerge the slice and was then transferred to the recording chamber of the A1R confocal microscope (Nikon) for fluorescence imaging. Recording chamber temperature was maintained at 28°C by a temperature controller (Harvard Instruments) throughout the recording experiment. A Nikon plan Fluor 10 × 0.3 W DIC N1 water immersion objective lens was used for fluorescence imaging.

For the measurement of exocytosis and vesicle recycling, at first 1 min of baseline fluorescence was recorded in gassed resting solution (abbreviated as RS) followed by depolarization with 25 mM KCl-containing RS solution (denoted as depolarization solution) for 1 min to elicit exocytosis. After depolarization with depolarization solution, the slices were repolarized for 1 min by adding 2 mM $Ca^{2+}$-containing RS to the chamber for vesicle retrieval and to return to baseline fluorescence as the vesicle re-acidifies. Each slice was stepped through three depolarization-recovery cycles (by adding depolarization solution followed by its replacement with RS solution). The depolarizing and recovery solutions (i.e., depolarization solution and RS, 2.5 ml each) were added manually to the holding chamber from one side using a dropper, and the other side suction was applied to remove the solutions. The responses were measured by making a rectangular region of interest after focusing the synaptic layer with an acquisition rate of 1 Hz at emission wavelength of 545 nm and excitation wavelength of 488 nm.

### Data analyses of SypHy recordings

Data were exported to Excel for further analysis and normalized by setting the fluorescence baseline to one arbitrary unit. Curve fitting was performed with Igor Pro in order to measure kinetics and amplitude of the synaptic response from each slice. A double-exponential curve was fitting best for both the depolarization and repolarization responses demonstrating fast and slow mechanisms of vesicle cycling in rod photoreceptor synapses. The fit results were obtained as A1 (amplitude 1 of the fast response), T1 (tau 1, time constant for the fast response), A2 (amplitude 2 of the slow response), and T2 (tau 2, time constant for the slow response) from each recording. The resulting values for amplitudes or time constants were averaged and compared between the MOG/CFA-injected and the CFA-injected control mice group.

Statistical analysis was performed using Origin pro software. Samples with a normal distribution (Shapiro–Wilk test) were compared using unpaired Student's *t*-test. Mann–Whitney *U*-test, a non-parametric test, was applied for comparing samples which were not normally distributed according to Shapiro–Wilk test.

### Miscellaneous methods

SDS–PAGE and Western blotting experiments were performed as previously described (Schmitz *et al*, 2000; Wahl *et al*, 2016; Eich *et al*, 2017). Protein determination of retinal samples dissolved in Laemmli SDS sample buffer: Mouse retinas were isolated 5 min post-mortem and dissolved in 100 μl hot (96°C) Laemmli sample buffer. From this retinal lysate, protein concentration was determined exactly as described (Dieckmann-Schuppert & Schnittler, 1997).

### Quantification of Western blot bands

Individual files were loaded in ImageJ NIH software, and the intensity of the Western blot bands was determined by measuring the gray values of the Western blot bands. In order to obtain band intensities, the area under the curve was determined by making a rectangular region of interest (ROI) covering the respective band. The same ROI was used for all other bands. Integrated gray values of the Western blot bands were obtained as area under the signal peaks using the gel analyses tools of ImageJ. All data were exported to Excel, normalized to the respective controls, as indicated, and plotted as bar graphs.

### Heterologous expression of pCASPR1mCherryN1/pmCherryN1 in HEK293 cells

For heterologous expression, HEK293 cells were transfected with pCASPR1mCherryN1 (experimental plasmid) and pmCherryN1 (control plasmid) cDNAs using standard procedures (calcium phosphate method). HEK293 cells were split and plated 1 day before transfection on sterile glass petri dishes ($\approx$9 cm diameter). After transfection ($\approx$5 μg plasmid DNA per petri dish), cells were incubated for 48 h at 37°C before being harvested. Cells were collected with a cell scraper and sedimented by centrifugation at $\approx$95 *g* (10 min, 4°C). Cells were washed twice with ice-cold PBS before they were processed for lysate preparation as described above in the ELISA experiments. Cell lysates were analyzed by Western blot using the antibodies indicated in the respective experiments. Successful heterologous expression was verified by immunofluorescence microscopy and by Western blot analyses with mouse monoclonal antibodies against mCherry (Fig 4B) and rabbit polyclonal antibody against CASPR1 (data not shown). Blood sera from MOG/CFA- and CFA-treated mice were used at a 1:100 dilution in Western blot analyses.

In Fig 4E, Western blot signals from MOG/CFA post-injection bloods samples probed with lysates from pCASPR1mCherryN1-overexpressing cells (Fig 4D, upper lane 15, 16; arrowhead) were quantified as described above and set to 100%. Signals from CFA post-injection blood samples probed with lysates from pCASPR1mCherryN1-overexpressing cells (Fig 4D, upper lane 13, 14) were normalized to this reference value (100%). Also, Western blot signals from CFA and MOG/CFA post-injection blood samples probed on pmCherryN1 overexpressing lysates (Fig 4D, upper lane 9–12) were referenced to it. All the pre-injection samples in Fig 4C

were used to subtract the background signals for the respective post-injection samples.

## Assessment of visual behavior in mice

Visual acuity of mice was determined by identifying the highest spatial frequency able to trigger optomotor responses. Visual behavior was tested using a virtual optomotor system (Prusky *et al*, 2004; Alam *et al*, 2015), consisting of a recording chamber build by computer screens, an elevated platform for positioning of the mouse, and a video camera placed over the platform for monitoring the mouse.

In brief, mice were tested in the first few hours of their daylight cycle (usually between 9:00 and 11:00 a.m.). The mouse to be measured was placed in the recording chamber and allowed to habituate for 10 min. During this time, the computer screens projected the same gray background illumination that was also shown on the monitors in between pattern projection. To start assessment of visual acuity, the mouse was placed on the platform. The video camera was calibrated on the platform size. As the mouse can freely move on the platform, position of mouse head was tracked by the examiner using a crosshair cursor superimposed on the live video image. X–Y position of the cursor also centered rotation of the virtual cylinder at the mouse's viewing position. The grating presentation was started according to the software provided with the optomotor setup (OptoMotry; CerebralMechanics, Lethbride, Alberta, Canada). When a grating was presented that was perceptible to the mouse, it stopped moving on the platform and followed the cylinder rotation with reflexive head movements. Tracking behavior was monitored by the examiner in a live video. If reflexive head movements followed cylinder rotation, the examiner judged that the mouse could see the grating (positive response). If the mouse on the other hand did not track the grating (negative response), it was judged that grating could not be perceived by the mouse. Both types of responses were feed-backed to the system by the examiner in real time and the software changed the spatial frequency of the grating accordingly until the highest special frequency perceived by the mouse was determined. If during time of recording the mouse slipped or jumped of the platform, recording was paused, the mouse was placed on the platform again, and recording was resumed. Results (visual acuity of each eye individually and the combined frequency threshold) of each measurement were noted. Depending on the mouse, one measurement normally lasted for 5–20 min. Each mouse was measured three times with at least 30 min break in between the measurements. During breaks, mice were returned to their housing cages with access to water and food *ad libitum*. Examiner was blinded to treatment of the mice.

## Statistical analyses

First, data were tested for normality using Shapiro–Wilk test using OriginPro Software (OriginLab Corporation). If comparing groups, all analyses were done assuming unequal variances. Unless noted otherwise, the two-tailed Wilcoxon–Mann–Whitney *U*-test (significance level $\alpha = 0.05$) was used to compare two groups of samples. Analysis was done without knowledge of sample identity (blinded analysis). The applied statistical test is online available from the webpage of the Saarland University

**The paper explained**

**Problem**

Optic neuritis is one of the first manifestations of multiple sclerosis (MS), a disease that is considered to exclusively target myelinated structures in the central nervous system. The primary events that lead to optic neuritis in MS are not well understood.

**Results**

We demonstrate that ribbon synapses in the myelin-free retina are targeted by an auto-reactive immune system even before morphologically visible alterations in the optic nerve have developed. We show that a CASPR1/CNTN1-containing adhesion complex is highly enriched at retinal ribbon synapses and is extremely sensitive to inflammatory changes in the early, pre-clinical phase of experimental auto-immune encephalomyelitis (EAE), a mouse model of MS with frequent optic nerve affection. The early synaptic changes in EAE retinas are caused by a rapid and massive auto-immune response directed against retinal proteins, including auto-antibodies against CASPR1, that lead to an enhanced recruitment and activation of a local complement system at retinal synapses. This occurs in parallel with impaired synaptic vesicle cycling at photoreceptor synapses and altered visual behavior before the onset of optic nerve demyelination.

**Impact**

Except for the retinal ganglion cells, the retina has not yet been considered as a primary target in MS or optic neuritis. Our findings demonstrate that ribbon synapses in the retina are early targets of an auto-reactive immune system even before morphological alterations are visible in the optic nerve and propose that synapse dysfunctions might contribute to the pathophysiology of optic neuritis in MS/EAE.

(https://ccb-compute2.cs.uni-saarland.de/wtest/ (Marx *et al*, 2016). All other statistical analyses were performed with OriginPro statistics software from OriginLab Corporation (Northampton, MA, USA). Data are presented as means ± SEM. Number of independent experiments (*N*) and number of analyzed images (*n*) are indicated in the respective experiments. In all cases, experiments were repeated at least three times. In ANOVA analyses, a *post hoc* Bonferroni correction was performed.

**Expanded View** for this article is available online.

## Acknowledgements

We thank Gabriele Kiefer for excellent technical assistance, Prof. Dr. Jens Rettig and Dr. Varsha Pattu (Institute for Physiology, CIPMM, Saarland University) for stimulating discussions, and Dr. Jutta Schmitz-Kraemer for critically reading the manuscript. The mouse rod opsin promotor construct was a kind gift of Dr. Thomas C. Südhof (Stanford University, USA); SypHy cDNA was a kind gift of Dr. Rafael Fernandez-Chacon (University of Sevilla, Spain). Work of the authors was supported from research grants from the Deutsche Forschungsgemeinschaft (FOR2289 and SFB894).

## Author contributions

FS, RD, and VF designed the study; FS, MD, KS, VF, and RD wrote the paper; MD, SN, KS, CF-T, RF, SKW, and AK performed experiments. All authors analyzed data.

## Conflict of interest

The authors declare that they have no conflict of interest.

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
