## [Review Process File · EMBO Molecular Medicine]

Early auto-immune targeting of photoreceptor ribbon synapses in mouse models of multiple sclerosis

Mayur Dembla, Ajay Kesharwani, Sivaraman Natarajan, Claudia Fecher-Trost, Richard Fairless, Sarah K. Williams, Veit Flockerzi, Ricarda Diem, Karin Schwarz, and Frank Schmitz

Review timeline:

Submission date:	28 January 2018
Editorial Decision:	19 March 2018
Revision received:	16 July 2018
Editorial Decision:	20 August 2018
Revision received:	31 August 2018
Accepted:	4 September 2018

Editor: Céline Carret

Transaction Report:

1st Editorial Decision

19 March 2018

Thank you for the submission of your manuscript to EMBO Molecular Medicine and your patience. We have now heard back from the three referees whom we asked to evaluate your manuscript.

You will see that while ref. 2 and 3 are encouraging and have defined suggestions to improve the study, ref. 1 is much more critical. Ref. 1 indeed finds the study to be descriptive and would like to see additional models to validate the concept, in terms of immunological, kinetic and electrophysiological analyses. Ref. 2 also misses some mechanism but suggests experiments to improve conclusiveness, present values and exact p-values and quantifications. This last part is a shared concern with ref. 3, who equally requests better co-localisation experiments, but also better discussion and explanations.

Furthermore, our cross-commenting exercise crystallised the important items to focus on: address issues as outlined by the reviewers' comments and focus on providing mechanistic details; key experiments would be: i) the localisation of CASPR/CNTN at the ribbon is new and should worked out further, as suggested by the reviewers. ii) showing a functional complement system at photoreceptor ribbons is a key point. Further, the authors should make sure that there is no onset of demyelination yet. Of course also all other comments should be addressed sufficiently.

We would welcome the submission of a revised version within three months for further consideration and would like to encourage you to address all the criticisms raised as suggested to improve conclusiveness and clarity. Please note that EMBO Molecular Medicine strongly supports a single round of revision and that, as acceptance or rejection of the manuscript will depend on another round of review, your responses should be as complete as possible.

I look forward to receiving your revised manuscript.

***** Reviewer's comments *****

Referee #1 (Comments on Novelty/Model System for Author):

ad 3.

The study is very descriptive and lacks a compelling direct and functional link between alterations of Caspr1 expression in the retina, the presence of complement components in the retina, anti-retinal antibodies in the serum and visuo-spatial behavior. This would be needed to present a "novel concept" with significant medical impact as mentioned in the title.

ad 4.

A T cell mediated MOG-model in mice alone is not sufficient to draw general conclusions on effects of immunization on retinal ribbon synapses. The relation of immunization with a T cell MOG peptide to short-term production of anti-retinal antibodies is unclear. Studies in other models e.g. MOG protein EAE, MOG genetic models or even models with different antigens used for immunization are advisable to exclude contaminations and prove the general validity of the concept.

Referee #1 (Remarks for Author):

In their manuscript „early immune targeting of photoreceptor ribbon synapses: a novel concept for optic neuritis" Dembla and co-workers co-immuno-purify CASPR1 and synaptic ribbons and use confocal microscopy to show that Caspr1 accumulates close to ribbon synapses. They then show alterations of synaptic Caspr1 expression in the retina as well as the presence of complement components day 9 after MOG immunization of mice. Finally, the authors show an increase in anti-retinal antibodies in the blood via ELISA and in a cell based assay and describe changes in spatial visual behavior.

The manuscript is well written and data technically look convincing. In general, the analysis of early retinal events in the pathogenesis of experimental neuroinflammation is of interest. Yet, there are several concerns with the present study:

1. As it stands, the study is very descriptive and lacks a compelling direct and functional link between alterations of Caspr1 expression in the retina, the presence of complement components in the retina, anti-retinal antibodies in the serum and visuo-spatial behavior. This would be needed to present a "novel concept" as mentioned in the title.
2. At many times, including the title, the authors try to generalize their finding to "optic neuritis". This is an overstatement. The authors only investigate MOG-EAE in mice as one model of MS with a very specific trigger which is probably more representative of anti-MOG disease as a new entity and not for MS itself or even optic neuritis alone.
3. As shown by some of the authors themselves, time kinetics of retinal damage significantly differ between different model strains (compare g. RCG loss between mice and rat EAE, Quinn et al, 2011 versus Meyer et al., 2001 or Hobom et al., 2004). Hence, a kinetic analysis over course of EAE is needed, actually showing the course of EAE and adaption processes in the retina over time - ribbon synapses are well known to exhibit plasticity depending on their functional activity. Would the authors expect similar findings in other EAE models?
4. The authors need to show that the presence of the complement system in photoreceptor ribbon synapses is really functionally active
5. The authors need to show that the antibodies in the serum are actually present in the retina and have pathogenic potential. How about retina specific B cell and plasma cell responses in their MOG-model? A more thorough immunological analysis is definitely needed.
6. Along this line, MOG peptide induced EAE is a T helper cell dominant EAE model with a clear (initial) epitope target. It is thus surprising to find a pivotal role of retina specific complement and antibodies which is apparently completely MOG independent that early in the disease. Do the

authors find T cell responses against retina or Caspr1 in their model? Is there evidence for early antigen spreading? How does a specific initial T cell response against MOG lead to changes in the retina without demyelination? Is MOG present anywhere in the retina? Or is there any contamination in their MOG synthesis? Do the authors have similar findings with different MOG preparations from other sources? Or also MBP as another myelin protein? If aiming at the presentation of a completely new concept, the authors definitely need to tackle this obvious paradox.

7. At several times, the authors imply their finding as first event before demyelination takes place. However, the study lacks data on the actual relation of synaptic changes to early signs of demyelination (e.g. in electron microscopy) in the same experiment. A kinetic analysis over time also investigating early demyelination with sensitive techniques is needed here.

8. Visuo-spatial behavior is a complex behavior certainly not only indicative of ribbon synapse function. The study lacks electrophysiology to prove definite functional synaptic alterations produced by complements and/or their antibodies.

9. The concept of early retinal alterations in MS is generally not new and already shown in patients with childhood MS or after a first demyelinating event (see e.g. Peres-Rico et al., 2014; Yeh et al., 2009). The authors should focus on their new findings on Caspr1 and ribbon synapses and tune down aspects of general novelty of their finding for the retina or MS.

In summary, the general conclusions drawn on a novel concept in optic neuritis are not backed by the data. The concept how immunization with a T cell MOG peptide leads to short-term production of functionally relevant anti-retina antibodies is unclear. Further studies are needed to present a compelling functional concept convincingly linking MOG immunization with anti-retina responses in the immune system, structural changes in ribbon synapses and functional outcome in mice.

Referee #2 (Remarks for Author):

Dembla et al. present an interesting study on the initiation of optic neuritis, which is an early manifestation of multiple sclerosis. They raised the hypothesis that the pathophysiology of optic neuritis already starts on the level of ribbon synapses in the myelin-free retina, in an early phase prior demyelination of the optic nerve. CASPR1 and CNTN1, usually found at axoglial junctions of myelinated axons, seem to be associated with RIBEYE at photoreceptor ribbon synapses as shown by immunopurification, mass spectrometry and immunofluorescence. The retina seems to be target of an auto-reactive immune system, and CASPR1 and CNTN1 were reduced at synaptic ribbons in a mouse model for experimental autoimmune encephalomyelitis (EAE), which can be induced via injection of myelin oligodendrocyte glycoprotein (MOG). The pre-existing complement system, as further shown in the study, is hereby strikingly reinforced in this mouse model.

In principle I favor the publication in EMBO molecular medicine, because the authors present a number of interesting novel findings. Further, they use a combination of several approaches to support their findings and finally can even present a mechanism for the onset of optic neuritis, which is in contrast to what has been proposed before and therefore potentially very interesting for a broad community.

Still, I have some concerns that should be taken into account:

Major

I am missing a summary of the key findings and the potential mechanism(s) involved, maybe even by a schematic drawing.

Results

Generally, some paragraphs are lacking some details, though, most information is provided in the figures. The authors should make sure that each figure and result is sufficiently presented in the result part. For example:

-The results of Supplement Figure 4 are not sufficiently integrated in the result part, it is barely mentioned there.

-Immunofluorescence of the ribbon and CASPR1/CNTN1 is very beautiful, still it does not clearly show where exactly the complex of CASPR1 and CNTN1 is found at the ribbon or in its vicinity:

I would suggest to co-stain with bassoon as a clear marker close to the membrane, which would help to get a clear orientation.

I wonder if some of the high-resolution images/close-ups should go into the main figures.

Further, as the authors claim that unfortunately CASPR1 antibodies seem not work on immunogold level. But how about CNTN1? Any ultrastructural investigation would be very supportive for the study.

-Please provide the values and exact p-values for the quantifications, maybe in tables?

-I would be interested to see an ultrastructural analysis of the MOG/CFA injected mice compared to the control situation: Can any changes in ribbon size be detected? This could additionally be determined on the immunofluorescence level.

Discussion

-Possible interaction mechanism of CASPR1/CNTN1 with RIBEYE or other components at the synapse?

Minor

In general the authors should go over the text and check for inconsistencies, for example sometimes between value and unit is a space, sometimes not.

Introduction

The introduction has a strange order. This certainly might be a matter of taste, but in my opinion the authors should first introduce the relevant information and at the end point out the motivation and approach of their study. Further, they should introduce the complement system as well as the EAE in the introduction already.

Results

-Please always be consistent with naming the control and the MOG-injected mice (I believe it should be MOG/CFA)

Figures

I prefer seeing the individual data points in the plots if any possible.

-Please provide a title for each figure

-In Figure 1A lane 3 is below the indicated RIBEYE/CNTN1 complex a relatively prominent double band. Can the authors comment on that?

-Figure 2: I would not mirror the images, to me this was a bit irritating, but also this might be a matter of taste.

-Figure 4C: Please be consistent with labeling: sometimes it is CFA (Control) sometimes the other way around (in Fig. 3C).

-Figure 5: I would switch A and B, to have MOG/CFA next to each other.

-Figure 5G: Looks a bit lonely, should be rearranged.

-Figure 6: Please use always the same font size: A and B are much smaller than C-F. This is anyway a point, please work on the figures, they could be better arranged, sometimes the font/labeling of the blots or gels is really small and depicted in different font sizes.

Referee #3 (Remarks for Author):

Optic neuritis is a common symptom of multiple sclerosis (MS) characterized by demyelination of the optic nerve carrying visual information from the retina to higher brain areas. Demyelination and damage to the optic nerve (comprised of retinal ganglion cell axons) is believed to be the first manifestation of the disease. In this article Dembla et al., use a mouse model of optic neuritis to characterize a novel effect on retinal ribbon synapse proteins before signs of any optic nerve demyelination. This finding revises our understanding of the pathophysiology of optic neuritis and reveals retinal alterations as the starting point of the disease.

The authors uncover interactions of the ribbon protein RIBEYE with CASPR1 and CNTN1 proteins; also localized at the photoreceptor synaptic terminals. The authors then show these ribbon associated proteins to be downregulated at an early time-point of optic neuritis/MS. Further the complement system is recruited to these synapses at this early time-point and auto-reactive antibodies against CASPR can be detected in blood samples from mice at a pre-clinical stage of optic neuritis. These synaptic alterations at the ribbon synapse also compromise visual acuity. This is a comprehensive study proving clear effects on ribbon synapse proteins (and function) at an early stage of optic neuritis. The manuscript is well-written and the data convincingly support the conclusions that are laid out. I strongly support the publication of this article after the following concerns are addressed:

1. Data from Figure 3, as quantified in Figure 3C show significant downregulation of RIBEYE, CASPR and CNTN in both the OPL and the IPL, indicating that ribbon synapses of the IPL are also affected at this early stage of optic neuritis. Yet the co-localization images (also with super-resolution microscopy) in the figures and supplemental data show localization only at OPL ribbons. Why is localization of these proteins at the level of IPL ribbons not shown? Super-resolution images depicting the spatial arrangements and relationships of these three proteins at IPL ribbons should also be included. Similar to using PSD95 to label photoreceptor terminals for OPL ribbons, a bipolar terminal marker can be used to label presynaptic bipolar terminals for IPL ribbon and for demonstrating the localization of these proteins at the level of the IPL.
2. Quantification of immunofluorescence data: Why is puncta number not quantified in OPL and IPL for the retinas from preclinical optic neuritis vs control for quantifying the downregulation of RIBEYE, CASPR, CNTN, and upregulation of the complement proteins? Through their figures the authors have demonstrated their ability to gather high resolution images of these retinal proteins at single puncta resolution and analyzing puncta number would convey to the reader the number (and proportion) of OPL ribbons and IPL ribbons affected in the early stage of optic neuritis. The fluorescence intensity measure is difficult to relate with the number of output (ribbon) synapses affected so the authors should quantify the number of puncta in the plexiform layers across conditions and include these bar graphs in the figures alongside the fluorescence intensity measure. It might be that the authors need to re-image sections to image the OPL and IPL separately to better discern the number of puncta in retinal sections. The same imaging parameters cannot be successfully applied for both OPL and IPL across conditions as OPL signals would saturate and IPL signals would not be able to be captured effectively. An example of this discord is already visible in Figure 2 as the RIBEYE signals in the IPL are not clearly visible in panels A2, B2, C2 and D2 even though RIBEYE for sure is present in the IPL and the authors do go on to quantify this signal in Figure 3. The reader needs to see the IPL signal of these proteins across conditions.
3. Discussion section: A discussion of the known effects of optic neuritis on retinal tissue is missing from the discussion section. It would be informative to the readers if the known alterations at the level of ganglion cells (together with the time-line in relation to the progression of the disease) is discussed before the authors discuss their current findings.
4. Figure 4D: Please quantify the western blot increase for the complement proteins similar to the quantification of the western blot data in Suppl Figure 11 showing the downregulation of RIBEYE and CASPR: the reader cannot visually gauge the significance of the complement upregulation shown in Figure 4D.
5. Suppl Figure 8: The experiments showing overlap of CASPR with RIM/CASK need to be quantified for the % of colocalization between these markers. Also a random overlap estimate generated by flipping one channel relative to the other can generate a random colocalization estimate and better convey the extent of colocalization between these markers.
6. Figure 4: Localization of Complement with respect to RIBEYE needs to be shown at a higher magnification at both the OPL and IPL, since changes are observed at both the plexiform layers (see 4C). Further, a co-label of the complement C3 with PSD95 at the OPL would be helpful to determine its localization at presynaptic photoreceptor terminals.
7. Similarly for Figure 5, images for the localization of Complement 5b-9 at IPL should be included and a co-label with the presynaptic (photoreceptor terminal for OPL, and bipolar terminal for IPL) marker should be included to verify the localization at ribbon presynapses.
8. Suppl Figure 11: Western blot analyses reveal clear reduction for RIBEYE and CASPR in retinas from animals with pre-clinical optic neuritis. What about total protein levels for CNTN? Were the total proteins levels of CNTN also checked with Western blots?
9. Suppl Figure 5C: Please show signals of CASPR + RIBEYE, CNTN + RIBEYE and CASPR + CNTN before showing the overlay of all three.
10. Suppl Figure 7: Please show the overlay of CASPR + PSD95 and RIBEYE + PSD95 before showing overlay of all three channels.

11. Suppl Figure 9: The expression pattern of CASPR at this early time point is not punctate like RIBEYE and this difference is interesting: why not try to co-label with developing early photoreceptors at the time-point to show definitively the spread of the CASPR signal within the developing photoreceptors?

12. Details of the scope/objectives used for confocal and SR-SIM are currently lacking in the methods section. Please include these details on the microscopy, imaging resolution etc. Also clearly state animal numbers in the figure legends: there are several "N" and "n" listed and it's not clear across legends which relate to animal numbers.

1st Revision - authors' response

16 July 2018

Reviewer #1

#1.) „Referee #1 (Comments on Novelty/Model System for Author):
ad 3.

The study is very descriptive and lacks a compelling direct and functional link between alterations of Caspr1 expression in the retina, the presence of complement components in the retina, anti-retinal antibodies in the serum and visuo-spatial behavior. This would be needed to present a "novel concept" with significant medical impact as mentioned in the title.

Response: We added the requested functional data in the revised version of the manuscript (presented in Fig. 6). For this purpose, we generated a new transgenic reporter mouse for the optical recording of synaptic activity, the SypHy mouse. The SypHy transgene is under the control of the mouse rod opsin promoter to drive expression in rod photoreceptor synapses. Rod photoreceptor synapses were chosen because these synapses were affected in all analyzed mouse models (Fig. 3C, Fig. 7C, Appendix Fig. S11, Appendix Fig. S15). We induced EAE in the SypHy transgenic reporter mice and demonstrate that rod photoreceptor synapses are functionally affected in the EAE mouse model of multiple sclerosis. We demonstrate a decrease in the amplitude of fast and slow exocytosis in MOG/CFA-induced EAE in comparison to CFA-injected control mice demonstrating a direct impact of EAE on synapse function.

#2.) „ad 4.

A T cell mediated MOG-model in mice alone is not sufficient to draw general conclusions on effects of immunization on retinal ribbon synapses. The relation of immunization with a T cell MOG peptide to short-term production of anti-retina antibodies is unclear. Studies in other models e.g. MOG protein EAE, MOG genetic models or even models with different antigens used for immunization are advisable to exclude contaminations and prove the general validity of the concept.

Response: We performed several experiments to address this concern of the reviewer. First, we analyzed EAE mice in which commercially available MOG/CFA suspensions (instead of home-made MOG/CFA suspensions) were used and obtained identical results as with the home-made MOG suspensions excluding the possibility that the observed effects depend on the source of the MOG/CFA suspension. Furthermore, we analyzed a second and independent mouse model of multiple sclerosis, in which EAE was induced by immunization with proteolipid protein (PLP) (Terry et al., 2016): Results were very similar (Fig. 7; Fig. EV2) to those obtained with MOG/complete Freund adjuvants (CFA) mice (using both self-made MOG suspensions and commercial MOG suspensions).

#3.) „Referee #1 (Remarks for Author):

In their manuscript „early immune targeting of photoreceptor ribbon synapses: a novel concept for optic neuritis" Dembla and co-workers co-immuno-purify CASPR1 and synaptic ribbons and use confocal microscopy to show that Caspr1 accumulates close to ribbon synapses. They then show alterations of synaptic Caspr1 expression in the retina as well as the presence of complement components day 9 after MOG immunization of mice. Finally, the authors show an increase in anti-retinal antibodies in the blood via ELISA and in a cell based assay and describe changes in spatial visual behavior.

The manuscript is well written and data technically look convincing. In general, the analysis of early retinal events in the pathogenesis of experimental neuroinflammation is of interest.“

Response. Thank you.

#4.) „1. As it stands, the study is very descriptive and lacks a compelling direct and functional link between alterations of Caspr1 expression in the retina, the presence of complement components in the retina, anti-retinal antibodies in the serum and visuo-spatial behavior. This would be needed to present a "novel concept" as mentioned in the title. „

Response: We added functional data, i.e. optical recording of synaptic activity using a newly generated transgenic reporter mouse. This mouse model showed a defect in exocytosis both for the fast and slow phase of release at photoreceptor synapses. These findings are in line with the decreased visuo-spatial behavior of the EAE mice (see pls. also our comment to criticism #1 above). We also toned down our writing. Although we presented very similar findings in different mouse models of multiple sclerosis, we are aware that the transfer to the human systems has to be further investigated.

#5.) „2. *At many times, including the title, the authors try to generalize their finding to "optic neuritis". This is an overstatement. The authors only investigate MOG-EAE in mice as one model of MS with a very specific trigger which is probably more representative of anti-MOG disease as a new entity and not for MS itself or even optic neuritis alone.*“

Response: In the revised version of the manuscript, we also analyzed a completely independent mouse model of multiple sclerosis, the PLP mouse model (Terry et al., 2016), and found very similar effects as in the MOG/CFA-induced mouse model, i.e. rapid auto-immune response against CASPR1 concomitant with a decrease in the synaptic expression without alterations in the optic nerve. These findings support convincingly that synaptic changes precede alterations in the optic nerve both in the MOG/CFA-induced as well as in the PLP-induced mouse model of multiple sclerosis/optic neuritis.

#6.) „3. *As shown by some of the authors themselves, time kinetics of retinal damage significantly differ between different model strains (compare g. RCG loss between mice and rat EAE, Quinn et al, 2011 versus Meyer et al., 2001 or Hobom et al., 2004). Hence, a kinetic analysis over course of EAE is needed, actually showing the course of EAE and adaption processes in the retina over time - ribbon synapses are well known to exhibit plasticity depending on their functional activity. Would the authors expect similar findings in other EAE models?*“

Response: Yes. In addition to the MOG/CFA-induced mouse model of MS, we also investigated the PLP-induced mouse model of MS/optic neuritis (Terry et al., 2016). Also in the PLP mouse model, we found very similar changes as in the MOG/CFA-induced mouse model. We observed rapid auto-immune response against CASPR1 concomitant with a decrease in the synaptic expression without alterations in the optic nerve.

#7.) „4. *The authors need to show that the presence of the complement system in photoreceptor ribbon synapses is really functionally active*“

Response: The antibody C5b-9 is detecting a neo-epitope that is only formed when the complement is activated. Therefore, the C5b-9 immuno-signals were used as indicators of complement activation. Despite the activation of the complement system at retinal synapses, the synapses are still alive, although impaired in their activity, as we showed with optical recording of the SypHy reporter mice. After induction of EAE with MOG/CFA, SypHy mice showed severe impairment of vesicle cycling in comparison to control-injected animals (Fig. 6).

#8.) „5. *The authors need to show that the antibodies in the serum are actually present in the retina and have pathogenic potential. How about retina specific B cell and plasma cell responses in their MOG-model? A more thorough immunological analysis is definitely needed.*“

Response: We demonstrated that anti-CASPR1 auto-antibodies are not only present in the blood of MOG/CFA-injected (but not in control (CFA alone) -injected) mice but also in the cerebrospinal

fluid (CSF) demonstrating that the anti-CASPR1 antibody passed the blood-brain-barrier and is indeed available to bind to synapses. After binding to the synapse, the synapse-targeted anti-CASPR1 auto-antibodies are eliciting the synaptic changes that we described at the morphological and functional level.

#9.) „6. Along this line, MOG peptide induced EAE is a T helper cell dominant EAE model with a clear (initial) epitope target. It is thus surprising to find a pivotal role of retina specific complement and antibodies which is apparently completely MOG independent that early in the disease. Do the authors find T cell responses against retina or Caspr1 in their model? Is there evidence for early antigen spreading? How does a specific initial T cell response against MOG lead to changes in the retina without demyelination? Is MOG present anywhere in the retina? Or is there any contamination in their MOG synthesis? Do the authors have similar findings with different MOG preparations from other sources? Or also MBP as another myelin protein? If aiming at the presentation of a completely new concept, the authors definitely need to tackle this obvious paradox.“

Response: In the revised version of the manuscript, we demonstrate that synaptic changes occur before alterations in the optic nerve are obvious (Figs. 5,7; Appendix Fig. S12). The absence of alterations in the optic nerve at the time when synaptic changes are already present was demonstrated by electron microscopic analyses (e.g. by determination of myelin thickness), axon number counts and also by qualitative and quantitative evaluation of the immunolabelled nodal and paranodal regions in the optic nerves that were sourced from the same mice from which synapse function was evaluated (Fig. 5, Fig. 7, Appendix Fig. S12). The paranodal and nodal zone are sensitive readouts for early dysfunctions of the optic nerve (Stojic et al., 2018). And these parameters of the optic nerve are completely normal in the mice that already displayed severe synaptic pathology. We showed in the morphological analyses at the light and electron microscopic level that CASPR1 and CNTN1 are highly enriched at retinal ribbon synapses. The rapid and early auto-immune response against CASPR1 will therefore target retinal synapses. The antibodies against CASPR1 are present in the CSF and therefore principally available to bind to synaptic components such as CASPR1 and CNTN1. The auto-immune response against CASPR1 both in the MOG/CFA as well as in the PLP mouse model is most likely generated by epitope spreading which is a common, though incompletely understood phenomenon in auto-immune diseases including multiple sclerosis (e.g. McMahon et al., 2005; Flytzani et al., 2015).

#10.) “7. At several times, the authors imply their finding as first event before demyelination takes place. However, the study lacks data on the actual relation of synaptic changes to early signs of demyelination (e.g. in electron microscopy) in the same experiment. A kinetic analysis over time also investigating early demyelination with sensitive techniques is needed here.“

Response: We showed the absence of demyelination as well as other parameters by electron microscopy, as requested, and also by additional immunolabelling analyses of the nodal and paranodal regions (Figs. 5,7; Appendix Fig. S12).

#11.) „8. Visuo-spatial behavior is a complex behavior certainly not only indicative of ribbon synapse function. The study lacks electrophysiology to prove definite functional synaptic alterations produced by complements and/or their antibodies.“

Response: We agree with the reviewer. Therefore we generated the new transgenic SypHy reporter mouse, as described in the manuscript, for optical recording of synaptic vesicle cycling. The analyses of this reporter demonstrated at a reasonable high temporal resolution changes both in the fast and slow phase of exocytosis together also with some changes in the kinetics of exocytosis. These novel data are presented in Fig. 6.

#12.) „9. The concept of early retinal alterations in MS is generally not new and already shown in patients with childhood MS or after a first demyelinating event (see e.g Peres-Rico et al., 2014; Yeh et al., 2009). The authors should focus on their new findings on Caspr1 and ribbon synapses and tune down aspects of general novelty of their finding for the retina or MS.“

Response. In the revised version of the manuscript, we included more literature references, e.g. optical coherence tomography (OCT) literature, that pointed to an involvement of the synaptic

layers in MS patients. Our new point is that these changes occur very early, i.e. before changes in the optic nerve and not after changes in the optic nerve as mentioned in the above cited papers. These findings are very novel and propose that synaptic changes could be responsible for the subsequent changes in the optic nerve. But we agree and tuned down speculations and re-wrote the manuscript accordingly.

„Referee #2 (Remarks for Author):

Dembla et al. present an interesting study on the initiation of optic neuritis, which is an early manifestation of multiple sclerosis. They raised the hypothesis that the pathophysiology of optic neuritis already starts on the level of ribbon synapses in the myelin-free retina, in an early phase prior demyelination of the optic nerve. CASPR1 and CNTN1, usually found at axoglial junctions of myelinated axons, seem to be associated with RIBEYE at photoreceptor ribbon synapses as shown by immunopurification, mass spectrometry and immunofluorescence. The retina seems to be target of an auto-reactive immune system, and CASPR1 and CNTN1 were reduced at synaptic ribbons in a mouse model for experimental autoimmune encephalomyelitis (EAE), which can be induced via injection of myelin oligodendrocyte glycoprotein (MOG). The pre-existing complement system, as further shown in the study, is hereby strikingly reinforced in this mouse model.

In principle I favor the publication in EMBO molecular medicine, because the authors present a number of interesting novel findings. Further, they use a combination of several approaches to support their findings and finally can even present a mechanism for the onset of optic neuritis, which is in contrast to what has been proposed before and therefore potentially very interesting for a broad community. „

Response: Thank you.

#13.) „I am missing a summary of the key findings and the potential mechanism(s) involved, maybe even by a schematic drawing.“

Response: In the revised version, we improved the writing of the key findings in the discussion. We also added a schematic drawing that summarizes our findings (graphic abstract, synopsis figure).

#14.) „Generally, some paragraphs are lacking some details, though, most information is provided in the figures. The authors should make sure that each figure and result is sufficiently presented in the result part. For example:

-The results of Supplement Figure 4 are not sufficiently integrated in the result part, it is barely mentioned there. „

Response: Integration of the figures was improved in the revised version of the manuscript. This also includes Supplementary Fig. 4. Supplementary Fig. 4 (now Appendix Fig. S4) was mentioned in the original version of the manuscript only in the Materials and Methods part. In general we tried to better incorporate the figures in the text flow.

#15.) „Immunofluorescence of the ribbon and CASPR1/CNTN1 is very beautiful, still it does not clearly show where exactly the complex of CASPR1 and CNTN1 is found at the ribbon or in its vicinity:

I would suggest to co-stain with bassoon as a clear marker close to the membrane, which would help to get a clear orientation.

I wonder if some of the high-resolution images/close-ups should go into the main figures.

Further, as the authors claim that unfortunately CASPR1 antibodies seem not work on immunogold level. But how about CNTN1? Any ultrastructural investigation would be very supportive for the study.“

Response: In the revised version of the manuscript, we provided the ultrastructural localization of CASPR1 and CNTN1 using a newly generated monoclonal antibody against CASPR1 and a commercially available antibody against CNTN1. We were quite happy that the ultrastructural localization of both CASPR1 and CNTN1 succeeded because the ultrastructural localization at the EM level gives the best level of resolution on the localization of the respective proteins. The ultrastructural localization of CASPR1 was resolved by pre-embedding immunogold electron

microscopy using ultrasmall gold particles and subsequent silver enhancement on cryostat sections of the mouse retina. The ultrastructural localization of CNTN1 was resolved by post-embedding immunogold electron microscopy. Both procedures on the two different proteins show a very similar ultrastructural distribution of the two proteins, i.e. an enrichment at the synaptic ribbon as well as at some other sites (presynaptic plasma membrane at the active zone and plasma membrane at the postsynaptic tips of horizontal cells directly opposite to the presynaptic active zone).

#16.) „Please provide the values and exact p-values for the quantifications, maybe in tables?“

Response: the exact p-values were delivered in the revised version of the manuscript, either in the figure itself (in the column diagrams) or, if no more space was available in the figure itself, the precise p-values were given in the figure legend.

#17.) „I would be interested to see an ultrastructural analysis of the MOG/CFA injected mice compared to the control situation: Can any changes in ribbon size be detected? This could additionally be determined on the immunofluorescence level.“

Response: Yes, there are changes at the synaptic ribbon. We obtained many data concerning this aspect by high resolution confocal microscopy, SR-SIM microscopy, electron microscopy on cross-sections and EM serial sections. We would like to present these findings in an independent study (Kesharwani et al., in preparation). This also includes changes in the synaptic ribbon size both in X-/Y- direction as well as in Z-direction. But the current study already contains 27 figures (7 main figures, 5 Extended View Figures and 15 Appendix Figures). Therefore, we think that such an ultrastructural analysis of the synaptic ribbon complex and the detailed measurement of ribbon dimensions is beyond the scope of the present manuscript. But, we promise to submit these data soon in an independent study (these data represent the PhD thesis work of Ajay Kesharwani).

#18) -Possible interaction mechanism of CASPR1/CNTN1 with RIBEYE or other components at the synapse?

Response: We discussed possible mechanisms in the revised version of the manuscript.

#19) „Minor

In general the authors should go over the text and check for inconsistencies, for example sometimes between value and unit is a space, sometimes not.

Introduction

The introduction has a strange order. This certainly might be a matter of taste, but in my opinion the authors should first introduce the relevant information and at the end point out the motivation and approach of their study. Further, they should introduce the complement system as well as the EAE in the introduction already.“

Response: We re-arranged the writing, as requested.

#20.) Results

-Please always be consistent with naming the control and the MOG-injected mice (I believe it should be MOG/CFA)

Response: The writing was unified.

#21). „Figures

I prefer seeing the individual data points in the plots if any possible.“

Response: We agree with the reviewer, but considering all the experiments we performed and included in the manuscript within the three months (plus an additional four weeks) which we got for revision, we simply run out of time also to change all the plots. But, still, as the plots stand, they contain all essential informations.

#22.) „-In Figure 1A lane 3 is below the indicated RIBEYE/CNTN1 complex a relatively prominent double band. Can the authors comment on that?“

Response: At this time, not all bands have been identified by mass spectrometry yet. The identity of the double band mentioned by the reviewer is not known to us. We will analyze these and other bands in that IP in future investigations. In the current manuscript, we focused on the CASPR1/CNTN1. But, we clearly agree with the reviewer that it is definitely worthwhile to investigate these further co-purifying proteins.

#23.) „-Figure 2: I would not mirror the images, to me this was a bit irritating, but also this might be a matter of taste.“

Response: In the revised version of the manuscript, we omitted the mirror-switched display in Figs. 2, 3 as well as in Appendix Fig. S13.

#24.) -Figure 4C: Please be consistent with labeling: sometimes it is CFA (Control) sometimes the other way around (in Fig. 3C).

Response: Improved in Fig. 3 in the revised version of the manuscript

#25.) „-Figure 5: I would switch A and B, to have MOG/CFA next to each other.“

Response: (Fig. 5, now EV4) Thanks. Switched as suggested.

#26.) „-Figure 5G: Looks a bit lonely, should be rearranged.“

Response: We placed Fig. 5G more in the center of Fig. 5 (now Fig EV4)

#27.) „-Figure 6: Please use always the same font size: A and B are much smaller than C-F. This is anyway a point, please work on the figures, they could be better arranged, sometimes the font/labeling of the blots or gels is really small and depicted in different font sizes.“

Response: font size in Fig 6 (now Fig. 4) were unified. Care was taken to depict the labelling in larger sizes.

28) „Referee #3 (Remarks for Author):

Optic neuritis is a common symptom of multiple sclerosis (MS) characterized by demyelination of the optic nerve carrying visual information from the retina to higher brain areas. Demyelination and damage to the optic nerve (comprised of retinal ganglion cell axons) is believed to be the first manifestation of the disease. In this article Dembla et al., use a mouse model of optic neuritis to characterize a novel effect on retinal ribbon synapse proteins before signs of any optic nerve demyelination. This finding revises our understanding of the pathophysiology of optic neuritis and reveals retinal alterations as the starting point of the disease.

The authors uncover interactions of the ribbon protein RIBEYE with CASPR1 and CNTN1 proteins; also localized at the photoreceptor synaptic terminals. The authors then show these ribbon associated proteins to be downregulated at an early time-point of optic neuritis/MS. Further the complement system is recruited to these synapses at this early time-point and auto-reactive antibodies against CASPR can be detected in blood samples from mice at a pre-clinical stage of optic neuritis. These synaptic alterations at the ribbon synapse also compromise visual acuity. This is a comprehensive study proving clear effects on ribbon synapse proteins (and function) at an early stage of optic neuritis. The manuscript is well-written and the data convincingly support the conclusions that are laid out.“

Response: Thank you.

#29.) „1. Data from Figure 3, as quantified in Figure 3C show significant downregulation of RIBEYE, CASPR and CNTN in both the OPL and the IPL, indicating that ribbon synapses of the IPL are also affected at this early stage of optic neuritis. Yet the co-localization images (also with super-resolution microscopy) in the figures and supplemental data show localization only at OPL ribbons. Why is localization of these protein at the level of IPL ribbons not shown? Super-resolution images

depicting the spatial arrangements and relationships of these three proteins at IPL ribbons should also be included. Similar to using PSD95 to label photoreceptor terminals for OPL ribbons, a bipolar terminal marker can be used to label presynaptic bipolar terminals for IPL ribbon and for demonstrating the localization of these proteins at the level of the IPL.“

Response: In the revised version of the manuscript, we provide quantification of CASPR1/CNTN1/C3/C5b puncta for both OPL and IPL (Appendix Fig. S15). Still, we focused on the OPL because also in the additional mouse model that we included in the revised version of the manuscript, the OPL was consistently affected by all markers while the changes in the IPL were more moderate, e.g. considering changes in RIBEYE. Furthermore, the IPL is much more heterogeneous and divergent than the OPL and require many more different markers. Since we already have 27 figures in the revised version of the manuscript we would like to shift the analyses of different types of ribbon synapses in the IPL to an independent study. We think that addressing different subtypes of ribbon synapses in the IPL is beyond the scope of the present manuscript. Changes on RIBEYE are complex and will be addressed in a separate study (please see also our comment to criticism #17).

#30.) „2. Quantification of immunofluorescence data: Why is puncta number not quantified in OPL and IPL for the retinas from preclinical optic neuritis vs control for quantifying the downregulation of RIBEYE, CASPR, CNTN, and upregulation of the complement proteins? Through their figures the authors have demonstrated their ability to gather high resolution images of these retinal proteins at single puncta resolution and analyzing puncta number would convey to the reader the number (and proportion) of OPL ribbons and IPL ribbons affected in the early stage of optic neuritis. The fluorescence intensity measure is difficult to relate with the number of output (ribbon) synapses affected so the authors should quantify the number of puncta in the plexiform layers across conditions and include these bar graphs in the figures alongside the fluorescence intensity measure. It might be that the authors need to re-image sections to image the OPL and IPL separately to better discern the number of puncta in retinal sections. The same imaging parameters cannot be successfully applied for both OPL and IPL across conditions as OPL signals would saturate and IPL signals would not be able to be captured effectively. An example of this discord is already visible in Figure 2 as the RIBEYE signals in the IPL are not clearly visible in panels A2, B2, C2 and D2 even though RIBEYE for sure is present in the IPL and the authors do go on to quantify this signal in Figure 3. The reader needs to see the IPL signal of these proteins across conditions.“

Response: Using the reviewer’s approach, we added quantification of xy puncta. We also obtained the ultrastructural localization of CASPR1 and CNTN1 in the OPL which are the most severely affected and sensitive synapses targeted by the auto-reactive immune system. Please also see our comment to criticism #29.

#31.) „3. Discussion section: A discussion of the known effects of optic neuritis on retinal tissue is missing from the discussion section. It would be informative to the readers if the known alterations at the level of ganglion cells (together with the time-line in relation to the progression of the disease) is discussed before the authors discuss their current findings.“

Response: Done as requested. A chapter was added to the discussion.

#32.) „4. Figure 4D: Please quantify the western blot increase for the complement proteins similar to the quantification of the western blot data in Suppl Figure 11 showing the downregulation of RIBEYE and CASPR: the reader cannot visually gauge the significance of the complement upregulation shown in Figure 4D.“

Response: As requested, quantification of the stain intensities of Western blots bands (C3 and C3c) was added to Fig. EV3 as Fig. EV3E in the revised version of the manuscript.

#33.) “5. Suppl Figure 8: The experiments showing overlap of CASPR with RIM/CASK need to be quantified for the % of colocalization between these markers. Also a random overlap estimate generated by flipping one channel relative to the other can generate a random colocalization estimate and better convey the extent of colocalization between these markers.“

Response: In the revised version of the manuscript, we obtained the ultrastructural (EM) localization of CASPR and CNTN1 by using pre-embedding immunogold electron microscopy using a newly in-house generated monoclonal antibody against CASPR and by using post-embedding immunogold electron microscopy using a commercially available antibody against CNTN1. The resolution of immunoelectron microscopy considerably extends beyond the level of light microscopical analyses. In agreement with the demonstrated light microscopical analyses, the EM analyses showed an enrichment at the synaptic ribbon, at the presynaptic plasma membrane close to the active zone (where also RIM is enriched) as well as at the dendritic tips of horizontal cells directly opposite to the active zone.

#34.) „8. *Suppl Figure 11: Western blot analyses reveal clear reduction for RIBEYE and CASPR in retinas from animals with pre-clinical optic neuritis. What about total protein levels for CNTN? Were the total proteins levels of CNTN also checked with Western blots?*“

Response: Yes, CNTN1 was also quantified by Western blot. The results were added to Figure EV5A; quantification in Fig. EV5b.

#35.) 9. *Suppl Figure 5C: Please show signals of CASPR + RIBEYE, CNTN + RIBEYE and CASPR + CNTN before showing the overlay of all three.*

Response: Done as requested (Please note: Suppl. Fig. 5 is Appendix Fig. S5 in the revised version of the manuscript). We also succeeded in obtaining the ultrastructural localization for both CASPR1 (pre-embedding immunogold electron microscopy) and CNTN1 (postembedding immunogold electron microscopy). Both antibodies show the same distribution, i.e. an enrichment of CASPR1/CNTN1 at the synaptic ribbon complex in addition to the pre-synaptic plasma membrane at the active zone and some immunosignal at the plasma membrane at the tips of postsynaptic horizontal cells directly opposite to the active zone. These novel ultrastructural findings and its consequences are also discussed in the revised version of the manuscript.

#36.) „10. *Suppl Figure 7: Please show the overlay of CASPR + PSD95 and RIBEYE + PSD95 before showing overlay of all three channels.*“

Response: Done as requested (Remark: Suppl Fig. 7 of the original manuscript is now Appendix Fig. S7 in the revised version of the manuscript).

#37.) „11. *Suppl Figure 9: The expression pattern of CASPR at this early time point is not punctate like RIBEYE and this difference is interesting: why not try to co-label with developing early photoreceptors at the time-point to show definitively the spread of the CASPR signal within the developing photoreceptors?*“

Response: We no longer included that figure in the revised version of the manuscript because we were able to resolve the precise ultrastructural localization of CASPR1 and CNTN1 (Fig. EV1) in the revised version of the manuscript. Therefore, we felt that this figure is no longer needed in the revised manuscript, also considering that the revised manuscript already has 27 figures. We agree: The analysis of CASPR1 in developing photoreceptors is interesting and could indicate that CASPR1 is already associated with immature synaptic ribbons at an early time point. We would like to address this interesting question in a future study. Suppl. Fig. 9 from the original version of the manuscript was removed from the revised version of the manuscript.

#38.) „12. *Details of the scope/objectives used for confocal and SR-SIM are currently lacking in the methods section. Please include these details on the microscopy, imaging resolution etc. Also clearly state animal numbers in the figure legends: there are several "N" and "n" listed and it's not clear across legends which relate to animal numbers.*“

Response: All details were added in the revised version of the manuscript. “N”s and “n”s were always given either in the figure itself or in the figure legend.

Remark: Literature references in the rebuttal letter are taken from the main manuscript; citation details are given in the manuscript text.

Thank you for the submission of your revised manuscript to EMBO Molecular Medicine. We have now received the enclosed reports from the referees that were asked to re-assess it. As you will see, while referees 2 and 3 are now supportive, referee 1 still has concerns.

Mainly, referee 1 questions the claim that "the pathophysiology of optic neuritis really starts with an auto-immune attack in the retina independently from the initial auto-antigen". After further consultation with the other referees, we arrived at the conclusion that should you wish to make that claim, you would have to perform the experiments listed by referee 1 to validate it. However, we would like to suggest instead that you tune-down and re-write some elements of the text to ensure that you do not focus on that claim.

Please make sure to provide a point-by-point letter to referee 1 and my comments regarding minor editorial changes.

***** Reviewer's comments *****

Referee #1 (Comments on Novelty/Model System for Author):

While realize that the authors have spend significant effort in revising the manuscript, some significant conceptual concerns remain. With the limited immunological insights at this stage, the rather provocative claim that the pathophysiology of optic neuritis really starts with an auto-immune attack in the retina independently from the initial auto-antigen is not sufficiently backed by the data.

Referee #1 (Remarks for Author):

I realize that the authors have spend significant effort in revising the manuscript. However, from an immunological point of view, pivotal questions of my previous critique still remain unanswered. Given the provocative hypothesis raised with the present study, I feel that some obvious inconsistencies still need to be addressed. In detail, MOG 35-55 peptide (and, in the revised version, also PLP)-induced EAE are primarily T helper cell mediated EAE models with defined myelin epitope targets which are not directly related to Caspr1/Cntn1 (and which are probably not present in the retina). It is thus surprising to find a pivotal role for a MOG independent, retina specific autoimmune response as early as 7 days after immunization.

To tackle this apparent paradox, the following points require Attention, best in kinetic analyses:

1. Do the authors find T cell responses against Caspr1/Cntn1 in their model?
2. Is their evidence for TCR cross-reactivity between Caspr1/Cntn1 and MOG (like MOG and neurofilament, see e.g. doi: 10.1038/nm.1975) or is there early antigen spreading?
3. Do the authors find respective B cell responses/plasma cell activation in their models?
4. What subclass and subtype do the anti-Caspr1/Cntn1 antibodies belong to (IgM versus IgG)?
5. Is the increase in anti-Caspr1/Cntn1 antibodies specific or part of an unspecific increase in immunoglobulins after immunization with MOG? Here, controls with CFA alone are not sufficient, but immunization with unspecific, non CNS peptides in CFA would be needed.
6. Do these antibodies actually bind in the retina? The newly presented data from the CSF are not sufficient to answer this question.
7. How can the authors exclude that remote functional changes in the optic nerve preceding demyelination govern the observed changes in retinal photoreceptor ribbon synapses and that their observation on an auto-immune response against the two adhesion proteins is just an epiphenomenon?

These points need to be substantiated to back the rather provocative claim that the pathophysiology of murine optic neuritis really starts MOG independently within the retina.

Referee #2 (Remarks for Author):

Dembla et al. included in the revised version of the manuscript a large number of new experiments in order to satisfy the reviewers' comments.

They included

1) the ultrastructural localization of CASPR1/CNTN1 at ribbon synapses using pre- and postembedding immunogold approaches. Both are clearly present at the ribbon synapse complex and the presynaptic plasma membrane.

2) experiments to show that upon early EAE ribbon synapses are already functional: to do so they used a newly generated transgenic SytHy reporter mouse, which can be used for optical recordings of synaptic activity. In the MOG-injected mice synapse function was impaired but synapses were still alive.

3) analysis of the optic nerve on myelination with immunofluorescence and electron microscopy, demonstrating that the myelin sheath was not altered at early time points, therefore the observed changes precede the typical signs of optic neuritis.

4) the investigation of an independent mouse model (PLP) for multiple sclerosis (MS) and accompanied optic neuritis with similar results to their main mouse model (MOG/CFA).

5) in addition to self-made suspensions of myelin oligodendrocyte glycoprotein (MOG)/CFA peptide they repeated the analysis on commercially available MOG suspensions and found for both suspensions a reduction of synaptic CASPR1 and CNTN1.

6) the auto-antibodies generated in early EAE (both in the MOG/CFA model as well as in the PLP model of MS/optic neuritis) are not only present in the blood but also present in the cerebrospinal fluid (CSF), i.e. beyond the blood-brain barrier, and thus available to target synapses in the CNS.

I very much appreciate these additional experiments, which strengthen the manuscript to a huge extent.

I am furthermore satisfied with the immunogold electron microscopy of both CASPR1 and CNTN1 on ribbon synapses, which I requested. The authors used pre-embedding for CASPR1 and post-embedding for CNTN1, which is absolutely fine. I appreciate that they got both antibodies (a new in-house made one for CASPR1) to work on the ultrastructural level. Now the localization at the ribbon synapse as well as the plasma membrane is convincing to me and strikingly strengthens the novel aspects of the manuscript, which is specifically that the retina/ribbon synapses is an early immune target preceding optic nerve neuritis.

The authors also worked on the figures and they improved a lot.

The new experiments strikingly strengthen their findings. I have no further concerns publishing the manuscript in EMBO molecular medicine.

Referee #3 (Remarks for Author):

The authors have addressed all the concerns raised by the reviewers and performed several additional confirmatory experiments which have substantially elevated the quality and the novelty of this study.

Please proof-read the manuscript prior to publication. This reviewer found the following minor corrections:

1. last sentence of the first introduction paragraph: please change "although affection of retinal layers" to "although effects on retinal layers"

2. appendix figure S10 legend third line: please change "but bot in" to "but not in"

3. please avoid using the term "significant" when discussing the immunogold localization of synaptic proteins. As no quantification was performed for this experiment using phrases such as

"significant portion of CASP1/CNTN1" is confusing to the reader as no percentages or metrics are associated with the discussion of this data.

2nd Revision - authors' response

31 August 2018

“Reviewer #1 (Remarks for Author):

While realize that the authors have spent significant effort in revising the manuscript...”

We thank the referee.

..., some significant conceptual concerns remain. With the limited immunological insights at this stage, the rather provocative claim that the pathophysiology of optic neuritis really starts with an auto-immune attack in the retina independently from the initial auto-antigen is not sufficiently backed by the data.

To tackle this apparent paradox, the following points require Attention, best in kinetic analyses:

- 1. Do the authors find T cell responses against Caspr1/Cntn1 in their model?*
- 2. Is their evidence for TCR cross-reactivity between Caspr1/Cntn1 and MOG (like MOG and neurofilament, see e.g. doi: 10.1038/nm.1975) or is there early antigen spreading?*
- 3. Do the authors find respective B cell responses/plasma cell activation in their models?*
- 4. What subclass and subtype do the anti-Caspr1/Cntn1 antibodies belong to (IgM versus IgG)?*
- 5. Is the increase in anti-Caspr1/Cntn1 antibodies specific or part of an unspecific increase in immunoglobulins after immunization with MOG? Here, controls with CFA alone are not sufficient, but immunization with unspecific, non CNS peptides in CFA would be needed.*
- 6. Do these antibodies actually bind in the retina? The newly presented data from the CSF are not sufficient to answer this question.*
- 7. How can the authors exclude that remote functional changes in the optic nerve preceding demyelination govern the observed changes in retinal photoreceptor ribbon synapses and that their observation on an auto-immune response against the two adhesion proteins is just an epiphenomenon?*

These points need to be substantiated to back the rather provocative claim that the pathophysiology of murine optic neuritis really starts MOG independently within the retina.”

The points by the referee refer to the immunological basis of our observation. We agree that these observations should be strengthened by additional studies to validate this claim. These studies will require a lengthy analysis, which we will pursue, but which is out of the scope of the current study, which already contains many data. We therefore tuned down our claim that "the pathophysiology of optic neuritis really starts with an auto-immune attack in the retina independently from the initial auto-antigen": We re-wrote corresponding text passages accordingly to make sure that we do not focus on that claim. In the revised version (version R2), we now only stated that the synaptic changes occurred before morphologically visible alterations in the optic nerve, as judged by our light- and electron microscopic analyses. We explicitly included in the manuscript the possibility that early functional changes in the optic nerve that might escape visibility in the applied morphological and electron microscopical analyses could influence the early synaptic changes (Discussion chapter, last sentence on page 11). We think that this is an important addition to the manuscript.

“Reviewer #2 (Remarks for Author):

Dembla et al. included in the revised version of the manuscript a large number of new experiments in order to satisfy the reviewers comments.

They included

- 1) the ultrastructural localization of CASP1/CNTN1 at ribbon synapses using pre- and postembedding immunogold approaches. Both are clearly present at the ribbon synapse complex and the presynaptic plasma membrane.*
- 2) experiments to show that upon early EAE ribbon synapses are already functional: to do so they used a newly generated transgenic SynHy reporter mouse, which can be used for optical recordings of synaptic activity. In the MOG injected mice synapse function was impaired but synapses were still*

alive.

3) analysis of the optical nerve on myelination with immunofluorescence and electron microscopy, demonstrating that the myelin sheath was not altered at early time points, therefore the observed changes precede the typical signs of optic neuritis

4) the investigation of an independent mouse model (PLP) for multiple sclerosis (MS) and accompanied optic neuritis with similar results their main mouse model (MOG/CFA)

5) in addition to self-made suspensions of myelin oligodendrocyte glycoprotein (MOG)/CFA peptide they repeated the analysis on a commercially available MOG suspensions and found for both suspensions a reduction of synaptic CASPR1 and CNTN1

6) the auto-antibodies generated in early EAE (both in the MOG/CFA model as well as in the PLP model of MS/optic neuritis) are not only present in the blood but also present in the cerebrospinal fluid (CSF), i.e. beyond the blood-brain-barrier, and thus available to target synapses in the CNS. I very much appreciate these additional experiments, which strengthen the manuscript to a huge extend.

I am furthermore satisfied with the immunogold electron microscopy of both CASPR1 and CNTN1 on ribbon synapses, which I requested. The authors used pre-embedding for CASPR1 and post-embedding for CNTN1, which is absolutely fine. I appreciate that they got both antibodies (a new in house made one for CASPR1) to work on the ultrastructural level. Now the localization at the ribbon synapse as well as the plasma membrane is convincing to me and strikingly strengthen the novel aspects of the manuscript, which is specifically that the retina/ribbon synapses is an early immune target preceding optic nerve neuritis.

The authors also worked on the figures and they improved a lot.

The new experiments strikingly strengthen their findings. I have no further concerns publishing the manuscript in EMBO molecular medicine.”

We thank the referee.

“Reviewer #3 (Remarks for Author):

The authors have addressed all the concerns raised by the reviewers and performed several additional confirmatory experiments which have substantially elevated the quality and the novelty of this study.”

We thank the referee.

“Please proof-read the manuscript prior to publication. This reviewer found the following minor corrections:

1. last sentence of the first introduction paragraph: please change "although affection of retinal layers" to "although effects on retinal layers" “

Has been changed.

“2. appendix figure S10 legend third line: please change "but bot in" to "but not in" “

Has been changed.

“3. please avoid using the term "significant" when discussing the immunogold localization of synaptic proteins. As no quantification was performed for this experiment using phrases such as "significant portion of CASPR1/CNTN1" is confusing to the reader as no percentages or metrics are associated with the discussion of this data.”

Has been changed.

Corresponding Author Name: Frank Schmitz, Mayur Dembla, Karin Schwarz

Manuscript Number: EMM-2018-08926